# Temporal transcriptomic dynamics in developing macaque neocortex

Longjiang Xu[1†], Zan Yuan[2,3†], Jiafeng Zhou[2,4†], Yuan Zhao[1†], Wei Liu[2,4], Shuaiyao Lu[1], Zhanlong He[1], Boqin Qiang[2,4], Pengcheng Shu[2,4,5]*, Yang Chen[2,4]*, Xiaozhong Peng[1,2,6,7]*

[1]Institute of Medical Biology Chinese Academy of Medical Sciences, Chinese Academy of Medical Science and Peking Union Medical College, Kunming, China; [2]Department of Molecular Biology and Biochemistry, Institute of Basic Medical Sciences, Medical Primate Research Center, Neuroscience Center, Chinese Academy of Medical Sciences, School of Basic Medicine Peking Union Medical College, Beijing, China; [3]Agricultural Bioinformatics Key Laboratory of Hubei Province, Hubei Engineering Technology Research Center of Agricultural Big Data, College of Informatics, Huazhong Agricultural University, Wuhan, China; [4]State Key Laboratory of Common Mechanism Research for Major Diseases, Chinese Academy of Medical Sciences, Peking Union Medical College, Beijing, China; [5]Chinese Institute for Brain Research, Beijing, China; [6]State Key Laboratory of Respiratory Health and Multimorbidity, Chinese Academy of Medical Sciences, Peking Union Medical College, Beijing, China; [7]Institute of Laboratory Animal Science, Chinese Academy of Medical Sciences & Peking Union Medical College, Beijing, China

*For correspondence:
pengcheng_shu@ibms.pumc.edu.cn (PS);
yc@ibms.pumc.edu.cn (YC);
pengxiaozhong@pumc.edu.cn (XP)

†These authors contributed equally to this work

Competing interest: The authors declare that no competing interests exist.

**Abstract** Despite intense research on mice, the transcriptional regulation of neocortical neurogenesis remains limited in humans and non-human primates. Cortical development in rhesus macaque is known to recapitulate multiple facets of cortical development in humans, including the complex composition of neural stem cells and the thicker supragranular layer. To characterize temporal shifts in transcriptomic programming responsible for differentiation from stem cells to neurons, we sampled parietal lobes of rhesus macaque at E40, E50, E70, E80, and E90, spanning the full period of prenatal neurogenesis. Single-cell RNA sequencing produced a transcriptomic atlas of developing parietal lobe in rhesus macaque neocortex. Identification of distinct cell types and neural stem cells emerging in different developmental stages revealed a terminally bifurcating trajectory from stem cells to neurons. Notably, deep-layer neurons appear in the early stages of neurogenesis, while upper-layer neurons appear later. While these different lineages show overlap in their differentiation program, cell fates are determined post-mitotically. Trajectories analysis from ventricular radial glia (vRGs) to outer radial glia (oRGs) revealed dynamic gene expression profiles and identified differential activation of *BMP*, *FGF*, and *WNT* signaling pathways between vRGs and oRGs. These results provide a comprehensive overview of the temporal patterns of gene expression leading to different fates of radial glial progenitors during neocortex layer formation.

## eLife assessment

This study presents a **useful** resource for the gene expression profiles of different cell types in the parietal lobe of the cerebral cortex of prenatal macaques. The evidence supporting the claims of the authors is **solid**, and revision has clarified some of the cell isolation and cell classification issues flagged by reviewers. This dataset will be of interest to developmental neurobiologists and could potentially be used for future comparative studies on early brain development.

## Introduction

The neocortex is the center for higher brain functions, such as perception and decision-making. Therefore, the dissection of its developmental processes can be informative of the mechanisms responsible for these functions. Several studies have advanced our understanding of the neocortical development principles in different species, especially in mice. Generally, the dorsal neocortex can be anatomically divided into six layers of cells occupied by distinct neuronal cell types. The deep-layer neurons project to the thalamus (layer VI neurons) and subcortical areas (layer V neurons), while neurons occupying more superficial layers (upper-layer neurons) preferentially form intracortical projections (*Jabaudon, 2017*). The generation of distinct excitatory neuron (EN) cell types follows a temporal pattern in which early-born neurons migrate to deep layers (i.e. layers V and VI), while the later-born neurons migrate and surpass early-born neurons to occupy the upper layers (layers II–IV) (*Rakic, 1974*). In *Drosophila*, several transcription factors (TFs) are sequentially expressed in neural stem cells to control the specification of daughter neurons, while very few such TFs have been identified in mammals thus far. Using single-cell RNA sequencing (scRNA-seq), Telley and colleagues found that daughter neurons exhibit the same transcriptional profiles of their respective progenitor radial glia (RG), although these apparently heritable expression patterns fade as neurons mature (*Telley et al., 2019*). However, the temporal expression profiles of neural stem cells and the contribution of these specific temporal expression patterns in determining neuronal fate have yet to be wholly clarified in humans and non-human primates (NHP). Over the years, NHP have been widely used in neuroscience research as mesoscale models of the human brain. Therefore, exploring the similarities and differences between NHP and human cortical neurogenesis could provide valuable insight into unique features during human neocortex development.

In mammals, RG cells are found in the ventricular zone (VZ), where they undergo proliferation and differentiation. The neocortex of primates exhibits an extra neurogenesis zone known as the outer subventricular zone (OSVZ), which is not present in rodents. As a result of evolution, the diversity of higher mammal cortical RG populations increases. Although ventricular radial glia (vRG) is also found in humans and NHP, the vast majority of RG in these higher species occupy the OSVZ and are therefore termed outer radial glia (oRG). oRG cells retain basal processes but lack apical junctions (*Hansen et al., 2010*) and divide in a process known as mitotic somal translocation, which differs from vRG (*Ostrem et al., 2014*). vRG and oRG are both accompanied by the expression of stem cell markers such as *PAX6* and exhibit extensive self-renewal and proliferative capacities (*Betizeau et al., 2013*). However, despite functional similarities, they have distinct molecular phenotypes. Previous scRNA-seq analyses have identified several molecular markers, including *HOPX* for oRGs, *CRYAB*, and *FBXO32* for vRGs (*Pollen et al., 2015*). Furthermore, oRGs are derived from vRGs, and vRGs exhibit obvious differences in numerous cell-extrinsic mechanisms, including activation of the *FGF-MAPK* cascade, *SHH*, *PTEN/AKT*, and *PDGF* pathways, and oxygen ($O_2$) levels. These pathways and factors involve three broad cellular processes: vRG maintenance, spindle orientation, and cell adhesion/extracellular matrix production (*Penisson et al., 2019*). Some TFs have been shown to participate in vRG generation, such as *INSM* and *TRNP1*. Moreover, the cell-intrinsic patterns of transcriptional regulation responsible for generating oRGs have not been fully characterized.

ScRNA-seq is a powerful tool for investigating developmental trajectories, defining cellular heterogeneity, and identifying novel cell subgroups (*Luecken and Theis, 2019*). Several groups have sampled prenatal mouse neocortex tissue for scRNA-seq (*Ruan et al., 2021*; *Di Bella et al., 2021*), as well as discrete, discontinuous prenatal developmental stages in humans and NHP (*Pollen et al., 2015*; *Trevino et al., 2021*; *Nowakowski et al., 2017*; *Eze et al., 2021*). The diversity and features of primate cortical progenitors have been explored (*Hansen et al., 2010*; *Betizeau et al., 2013*; *Pollen et al., 2015*; *Pebworth et al., 2021*). The temporally divergent regulatory mechanisms that govern cortical neuronal diversification at the early post-mitotic stage have also been focused on *Yuan et al., 2022*. However, studies spanning the full embryonic neurogenic stage in the neocortex of humans and other primates are still lacking. Rhesus macaque and humans share multiple aspects of neurogenesis, and more importantly, the rhesus monkey and human brains share more similar gene expression patterns than the brains of mice and humans (*Bakken et al., 2016*; *Bernard et al., 2012*; *Zeng et al., 2012*). To establish a comprehensive, global picture of the neurogenic processes in the rhesus macaque neocortex, which can be informative of neocortex evolution in NHP, we sampled neocortical tissue at five developmental stages (E40, E50, E70, E80, and E90) in rhesus macaque embryos,

spanning the full neurogenesis. Through strict quality control, cell-type annotation, and lineage trajectory inference, we identified two broad transcriptomic programs responsible for the differentiation of deep-layer and upper-layer neurons. We also defined the temporal expression patterns of neural stem cells, including oRGs, vRGs, and intermediate progenitors, and identified novel TFs involved in oRG generation. These findings substantially enhanced our understanding the development and evolution of the neocortex in primates.

## Results

### scRNA-seq analysis of cell types in the developing macaque neocortex

In order to establish a comprehensive view of the cellular composition of the rhesus macaque brain at different prenatal stages, we conducted scRNA-seq on the dissected parietal lobes of eight total rhesus macaque embryos, spanning five developmental stages of prenatal neurogenesis, including E40 (stage of peak neurogenesis) and E50 (Layer 6 formation); as well as at E70 (Layer 5 formation), E80 (Layer 4 formation), and E90 (Layer 2–3 formation) (*Figure 1A*, *Figure 1—figure supplement 1*; *Rakic, 1974*). We obtained a transcriptomic atlas from 53,259 cells after filtering out low-quality cells and removing potential doublets. Each embedding was visualized using uniform manifold approximation and projection (UMAP) of dimension reduction using Seurat, which identified 28 distinct cell clusters. All cell clusters present in the samples were then annotated into cell types (*Figure 1B and C*) based on their expression of molecular markers (*Figure 1D* and *Figure 1—figure supplement 2B to C*). According to the expression of the marker genes, we assigned identities of cell types to clusters including radial glia [RG], outer radial glia [oRG], intermediate progenitor cells [IPCs], ventral precursor cells [VP], excitatory neurons [ENs], inhibitory neurons [INs], glia intermediate precursor cells [gIPC], oligodendrocytes, astrocytes, ventral LGE-derived interneuron precursors and Cajal-Retzius cells, or non-neuronal cell types (including microglia, endothelial, meningeal/VALC [vascular cell]/pericyte and blood cells). Based on the expression of the marker genes, cluster 23 was identified as thalamic cells, which are small numbers of non-cortical cells captured in the sample collection at earlier time points. Each cell cluster was composed of multiple embryo samples, and the samples from similar stages generally harbored similar distributions of cell types.

In general, vRG (cluster 10) showed characteristic *VIM* and *PAX6* expression; oRG (cluster 12 and cluster 14) highly expressed *HOPX*, previously verified marker (*Pollen et al., 2015*); two clusters (cluster 8 and cluster 22) of IPCs that strongly expressed *EOMES*, and one cluster of IPCs (cluster 22) were all topologically close to RG, while the other IPC cluster (cluster 8) was closer to neurons, indicating the presence of two stages of IPCs (*Hevner, 2019*); ENs were identified by the expression of well-established markers, such as *NEUROD2* and *NEUROD6*; and the astrocyte and oligodendrocyte lineages were identified by *AQP4* and *SOX10* expression, respectively (*Silbereis et al., 2010*). In addition, we identified *DLX1* and *GAD*-positive cells, which suggested the presence of INs (*Figure 1C* and *Figure 1—figure supplement 2*).

Collectively, these results suggested that cortical neural progenitors undergo neurogenesis processes during the early stages of macaque prenatal cortical development, while gliogenic differentiation, including oligodendrocytes and astrocytes, occurs in later stages (*Figure 1A and B*).

### Distinct excitatory neuronal types sequentially emerge in developing cortex

To better understand the temporal dynamics of EN development and differentiation, we compiled a subset of all ENs and re-clustered them into 10 subclusters (EN1–10) (*Figure 2A*). We then calculated the relative expression levels of cellular markers for each subclusters and annotated the EN subclusters based on published descriptions of marker function (*Figure 2—figure supplement 1B*). We found that all EN subclusters could be well distinguished by differential expression of either deep-layer neuron markers (*BCL11B*, *FEZF2*, and *SOX5*) (*Tsyporin et al., 2021*) or upper-layer neuron markers (*CUX1* and *SATB2*) (*Li et al., 2020*; *He et al., 2017*; *Figure 2B and C*). In previous seminal studies, 3H-thymidine tracing in macaque rhesus unraveled the sequential generation of cortical neurons, with deep-layer neurons appearing prior to upper-layer neurons (*Rakic, 1974*). In agreement with this early work, we found that early-born neurons (E40 and E50) predominantly outnumbered later-born

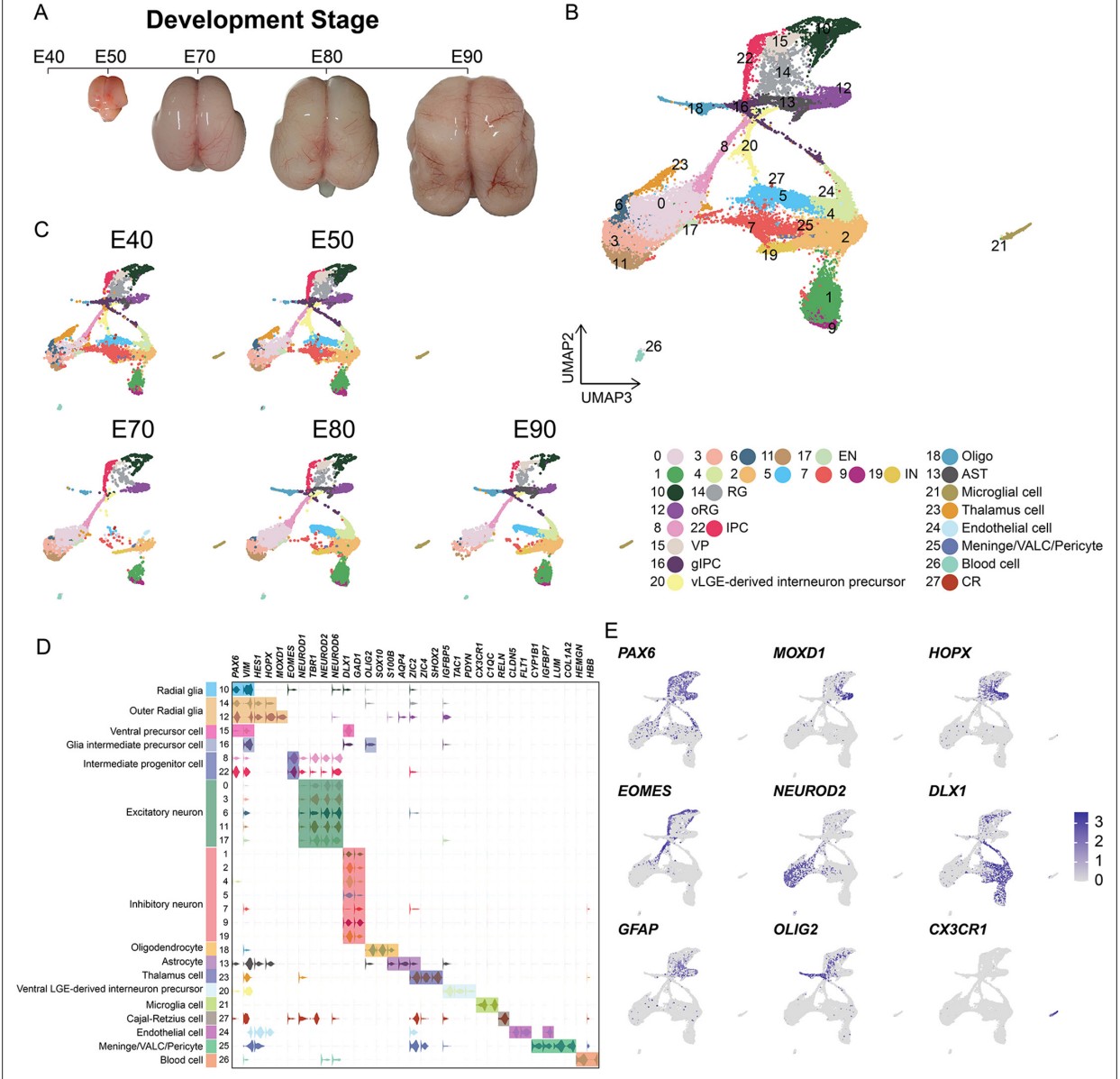

**Figure 1.** Cell types in macaque prenatal and fetal brain development. (**A**) Schematic diagram of sample collecting and data analysis. We collected the parietal lobe from the embryos across developmental stages from E40 to E90. (**B** and **C**) The transcriptome data of single cells were collected and used to do clustering using Seurat. Visualization of major types of cells using uniform manifold approximation and projection (UMAP). Dots, individual cells; color, clusters. (**D**) Violin plot of molecular markers for annotating cell types. (**E**) The expressions of the classic marker genes for each cell type were plotted for UMAP visualization. Light gray, no expression; dark blue, relative expression.

The online version of this article includes the following figure supplement(s) for figure 1:

**Figure supplement 1.** Sample collection and quality control.

**Figure supplement 2.** Single-cell RNA sequencing (scRNA-seq) uncovers cell type in the developing macaque neocortex.

neurons in the deep-layer neuron subclusters (EN5 and E10), while upper-layer neuron subclusters contained the inverse proportions (*Figure 2C and D*).

We then characterized temporal changes in the composition of each EN subcluster. While the EN5 and EN10 (deep-layer neurons) subclusters emerged at E40 and E50 and disappeared in later stages, EN subclusters 1, 2, 3, and 4 gradually increased in population size from E50 to E80, EN subclusters 8 and 9 gradually increased in population size from E80 to E90 (*Figure 2D*). Notably, EN6 was

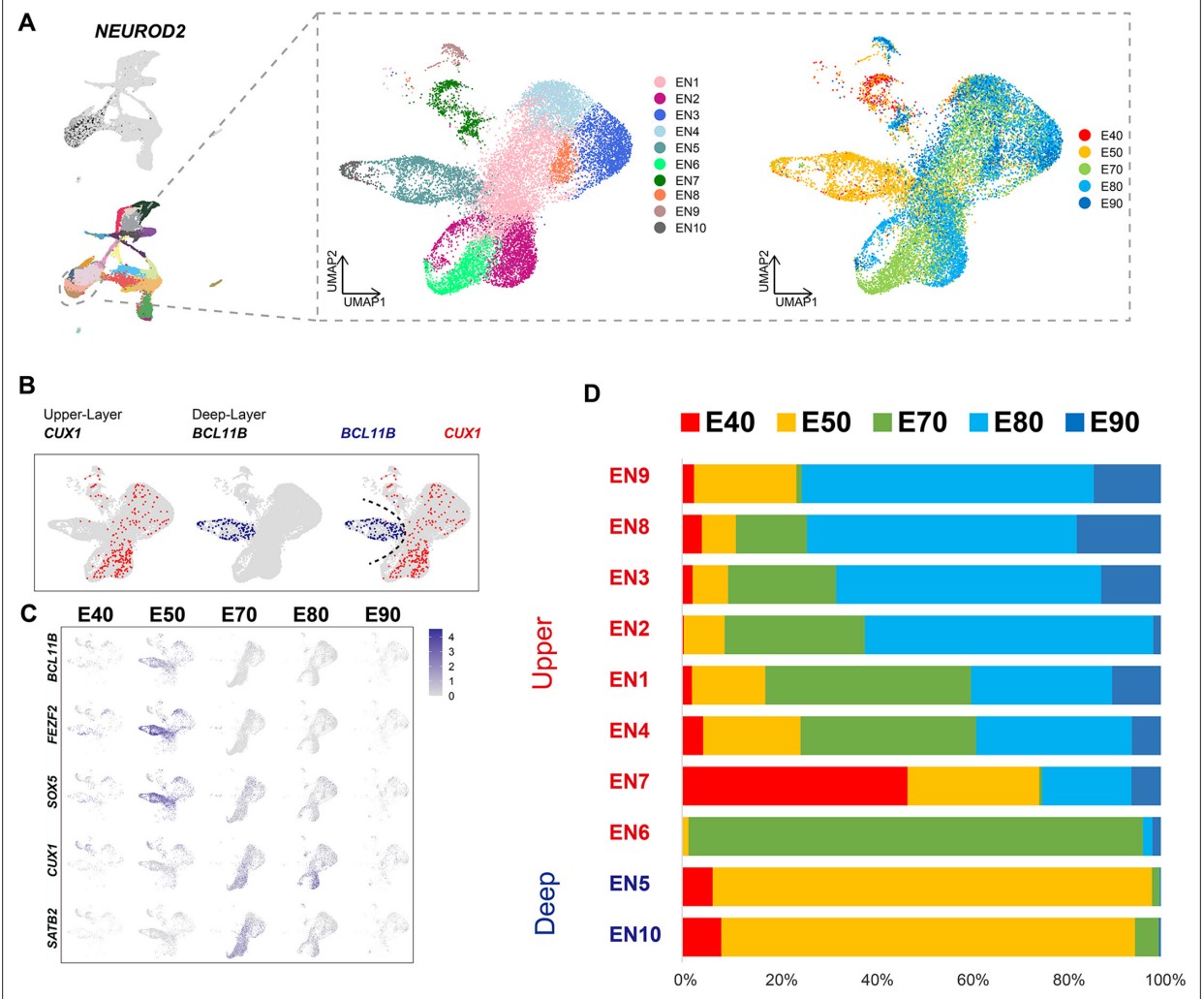

**Figure 2.** Excitatory neuron subclusters in the developing macaque cerebral cortex. (**A**) Left, clustering of excitatory neuron subclusters collected at all time points, visualized via uniform manifold approximation and projection (UMAP). Cells are colored according to subcluster identities (left) and collection time points (right). (**B**) Differentially, the expression of deep-layer marker *BCL11B* and upper-layer marker *CUX1* are highlighted. (**C**) Excitatory neuron subclusters' UMAP plot shows the expression of classic markers for deep layers (*BCL11B*, *FEZF2*, *SOX5*) and upper layers (*CUX1*, *SATB2*) present at each time point. (**D**) The proportion of different excitatory neuron subclusters corresponding to excitatory neurons in each time point.

The online version of this article includes the following figure supplement(s) for figure 2:

**Figure supplement 1.** Additional information for excitatory neuron subcluster.

exclusively found in the cortex at E70 (*Figure 2—figure supplement 1A*). EN7 is identified as *CUX1*-positive, *PBX3*-positive, and *ZFHX3*-positive EN subcluster.

## Specification of different progenitor fates controlled by regulatory genes

To focus on the differentiation of neural progenitors, we subsetted cell clusters 8, 10, 12, 14, 15, 16, and 22, then annotated them as vRG (RG_C10), ORG (oRG_C12 and oRG_C14), IPCs (IPC_C22 and IPC_C8), VP (VP_C15), or gIPCs (gIPC_C16) (*Figure 3A*). Subclustering analysis revealed that these neural progenitors differentiated into diverse cell types via distinct trajectories across the macaque prenatal neocortex (*Figure 3B*). We next investigated trajectories within the neural stem cell pool. The oRG and IPC groups exhibited characteristically high specific expression of *HOPX* and *EOMES*, respectively.

Although the presence of oRGs is a well-established feature of primate neurogenesis, and their molecular markers are widely used, the genetic basis and molecular processes leading to their

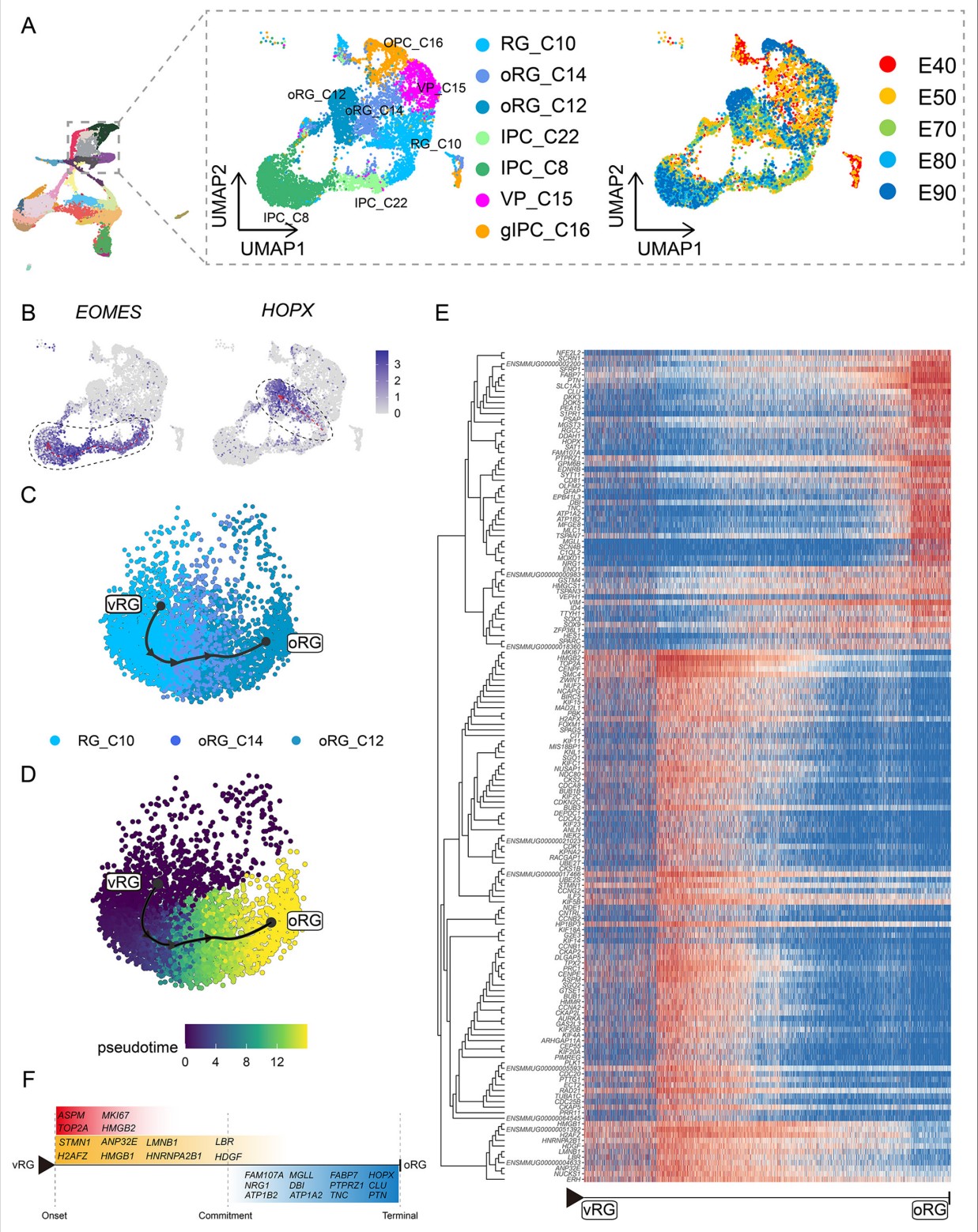

**Figure 3.** Cell diversity and regulation of progenitor cells in the macaque cortical neurogenesis. (**A**) Uniform manifold approximation and projection (UMAP) shows eight progenitor clusters and cell annotation. Left cells are colored according to Seurat clusters; right, cells are colored according to the collection time point. (**B**) Feature plot of outer radial glia (oRG) marker genes *HOPX* shows higher expression in C10–C14–C12 (left). Feature plots of intermediate progenitor cell marker gene *EOMES* show higher expression in C10–C22–C8 (right). (**C** and **D**) Pseudotime analysis by Slingshot of *HOPX*-positive cells (C10–C14–C12). The Slingshot result with the lines indicating the trajectories of lineages and the arrows indicating directions of

*Figure 3 continued on next page*

*Figure 3 continued*

the pseudotime. Cells are colored according to cell (**C**) and pesudotime (**D**). Dots: single cells; colors: cluster and subcluster identity. (**E**) The heatmap shows the relative expression of top 150 genes displaying significant changes along the pseudotime axis of radial glia (RG) to oRG (C10–C14–C12). The columns represent the cells being ordered along the pseudotime axis. (**F**) Schematic diagram of some significant genes related to (**E**). (The depth of the color indicates the levels of gene expression.)

The online version of this article includes the following figure supplement(s) for figure 3:

**Figure supplement 1.** Developmental regulation of gene expression from radial glia (RG) to intermediate progenitor cell (IPC).

**Figure supplement 2.** Transcriptional regulation of glia intermediate precursor cell (gIPC) differentiation into astrocytes and oligodendrocytes.

emergence are still poorly understood. To construct gene expression profiles that illustrate the progression from vRGs to oRGs, we specifically examined changes in expression among RG_C10, oRG_C12, and oRG_C14 cells, then calculated pseudotime trajectories (*Figure 3C to D*). Based on these trajectories, we then profiled the temporal shifts in the expression of each gene and selected the 150 most significant genes (*Figure 3E*). We found a set of genes that were previously reported as highly expressed in ORG that also showed high expression in the oRG_C12 differentiated pseudotime terminal (*Pollen et al., 2015*; *Liu et al., 2017*; *Johnson et al., 2018*) (such as *SFRP1*, *HOPX*, *FAM107A*, *TNC*, *PTN*, and *MOXD1*), while high *ASPM* expression was typical of cells in the earlier, RG, pseudo-time terminal (*Figure 3F*). In addition to these markers, we also found enrichment for some potential regulatory genes at the oRG_C12 terminal, such as regulator of the cell cycle (*RGCC*), which controls mitotic spindle orientation (*Guo et al., 2021*), and *TTYH1*, which regulates cell adhesion (*Zhou et al., 2020*). We also found genes that regulate excitability and ionic gradients in neurons, such as *ATP1A2*, *ATP1B2*, and *SCN4B*, which were not previously known to participate in oRG differentiation.

Analysis of DEGs specific to the RG pseudotime terminal identified cell cycle-related genes such as *MKI67*, *TOP2A*, *CDC20*, and *CCNA2*. Within the glia-like cells that largely comprised the oRG_C12 terminal, we also detected a population of glial-specific genes that exhibited high expression such as *SLC1A3*, *ZFP36L1* (*Weng et al., 2019*), *GFAP*, *DBI*, and *EDNRB* (*Gadea et al., 2008*). In addition, our analyses uncovered several DEGs that have not yet been investigated in RG function and differentiation, such as *DKK3*, *DDAH1*, *SAT1*, and *PEA15*. Finally, we screened significant DEGs associated with RG-to-IPC and gIPC-to-astrocyte/oligodendrocyte differentiation trajectories (*Figure 3—figure supplement 1* and *Figure 3—figure supplement 2*).

## The generation of deep-layer and upper-layer neurons follows distinct terminal trajectories

Based on our above finding that the deep-layer neurons appear earlier than upper-layer neurons, we next sought to characterize dynamic shifts in the expression of transcriptional regulators that potentially contribute to determining neuronal fates. To this end, we performed pseudotime trajectory analysis of EN population (including deep-layer and upper-layer neurons) (*Figure 4A*), then superimposed the cluster labels from RG_C10, oRG_C12, and oRG_C14 stem cells in UMAP plots (*Figure 3A*) and early (E40 and E50) and late (E70, E80, and E90) emergence stages (*Figure 2A*) over the extracted transcriptomic data. We then calculated the RG-specific differentially expressed genes (rgDEGs) and the DEGs of early and late neuron-specific (nDEGs). The set of genes that overlapped between rgDEGs and nDEGs (termed mapping genes) can thus be used to map neuronal subtypes to different neural stem cell populations (*Figure 4—figure supplement 1*). We hypothesized that mapping genes, such as *FEZF2* and *DOK5*, may play a role in RG cells to specify neuronal progeny (*Figure 4B*). Our results provide single-cell transcriptome data to support that apical progenitors and daughter ENs share molecular temporal identities during neocortex prenatal corticogenesis (; *Lin et al., 2021*; *Shu et al., 2019*).

Since substantial evidence indicates that neuronal fate is determined post-mitotically in the mammalian neocortex, analysis of DEGs throughout the differentiation process could reveal the genetic mechanisms responsible for deciding stem cell fate as different neuronal progeny. Using Dynverse R packages, we constructed pseudotime trajectories for the cluster data, which identified a bifurcating trajectory from neural stem cells (including RG_C10, oRG_C12, and oRG_C14) that leads to either deep-layer neurons in one branch or to upper-layer neurons in the alternative branch (*Figure 4C*). However, it should be noted that the two distinct fates share a common path from neural stem cells

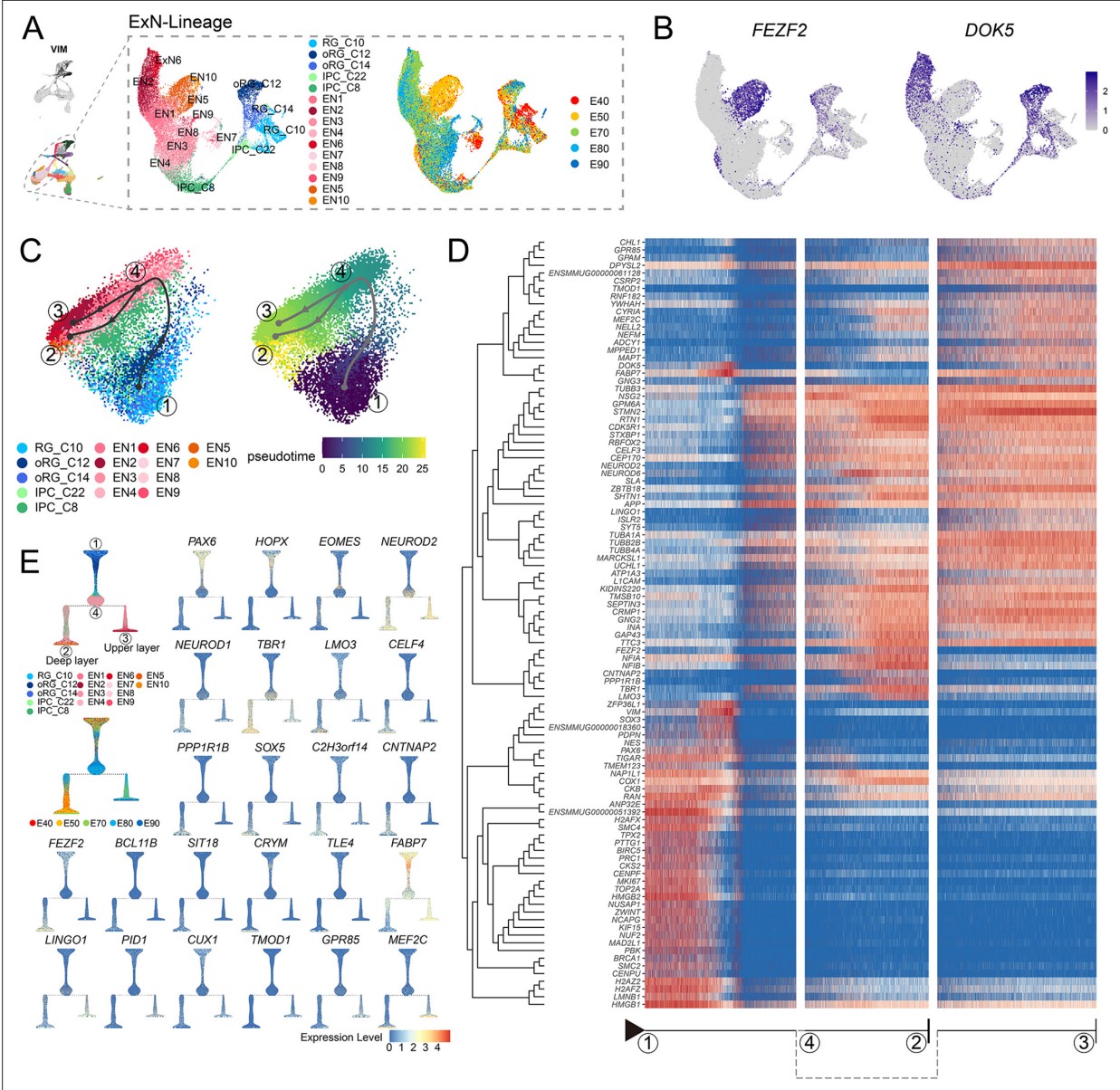

**Figure 4.** Transcriptional regulation of excitatory neuron lineage during prenatal cortical neurogenesis. (**A**) Uniform manifold approximation and projection (UMAP) shows the alignment of macaque cortical NPCs, intermediate progenitor cells (IPCs), and excitatory neurons. Left, cells are colored according to cell annotation. Different yellow/orange colors are used for deep-layer excitatory neuron subclusters (EN5 and EN10), and different red/pink colors are used for upper-layer excitatory neuron subclusters (EN1, EN2, EN3, EN4, EN6, EN7, EN9, and EN10). Right, cells are colored according to the time point of collection. (**B**) Dot plot showing the marker genes for the deep-layer excitatory neuron (*FEZF2*) and upper-layer excitatory neuron (*DOK5*). Light gray, no expression; dark blue, relative expression. (**C**) Pseudotime analysis by Slingshot projected on PCA plot of RGCs, oRGCs, IPCs, and excitatory neuron subclusters. The Slingshot result indicates the trajectories of lineages, and the arrows indicate the directions of the pseudotime. Dots: single cell; colors: cluster and subcluster identity. Framed numbers marked the start point, endpoint, and essential nodes of the Slingshot inference trajectory. Framed number '1' was the excitatory neuron lineage trajectory start point (C10). Framed number '4' marked immature neurons. Framed numbers '2' and '3' marked deep-layer and upper-layer neurons. Cells are colored according to cell annotation and pseudotime. (**D**) The heatmap shows the relative expression of the top 100 genes displaying significant changes along the pseudotime axis of each lineage branch. The columns represent the cells being ordered along the pseudotime axis. (**E**) Left, Slingshot branching tree related to Slingshot pseudotime analysis in C. The root is E40 earliest RG (C10), tips are deep-layer excitatory neurons generated at the early stage (E40, E50), and upper-layer excitatory neurons are generated at the later stage (E70, E80, E90). Right, branching trees showing the expression of marker genes of apical progenitors (*PAX6*), outer radial glia cells (*HOPX*), intermediate progenitors (*EOMES*), and excitatory neurons (*NEUROD2*), including callosal neurons (*SATB2, CUX2*), deeper layer neurons (*SOX5, FEZF2*), corticofugal neurons (*FEZF2, TLE4*). There is a sequential progression of radial glia cells, intermediate progenitors, and excitatory neurons.

The online version of this article includes the following figure supplement(s) for figure 4:

**Figure supplement 1.** Stem and excitatory neuron subcluster mapping genes.

to immature neurons (*Figure 4C*, from dot 1 to dot 4), supporting the likelihood that neuronal fate is primarily determined post-mitotically.

Based on this trajectory, we categorized the DEGs that exhibit dynamic, temporal shifts in expression over pseudotime. The first of these clusters was enriched with neural stem cells, characterized by high expression of *PAX6*, *VIM*, and *TOP2A*, which likely participate in regulating stemness and proliferation. The three remaining DEG clusters were enriched in either deep-layer neurons, upper-layer neurons, or both (*Figure 4D*). More specifically, we found that most DEGs, such as *STMN2*, *TUBB3*, *NEUROD2*, and *NEUROD6*, were shared between both branches, illustrating the common differentiation processes between deep- and upper-layer ENs. We also identified deep-layer-specific DEGs (including *FEZF2*, *TBR1*, and *LMO3*) and upper-layer-specific genes (including *MEF2C* and *DOK5*). We then organized the cell types into a branching tree based on their differential expression of these marker genes, including stem cell genes *PAX6*, *HOPX*, and *EOMES*, in addition to the above-mentioned deep-layer- and upper-layer-specific genes (*Figure 4E*). The separation into

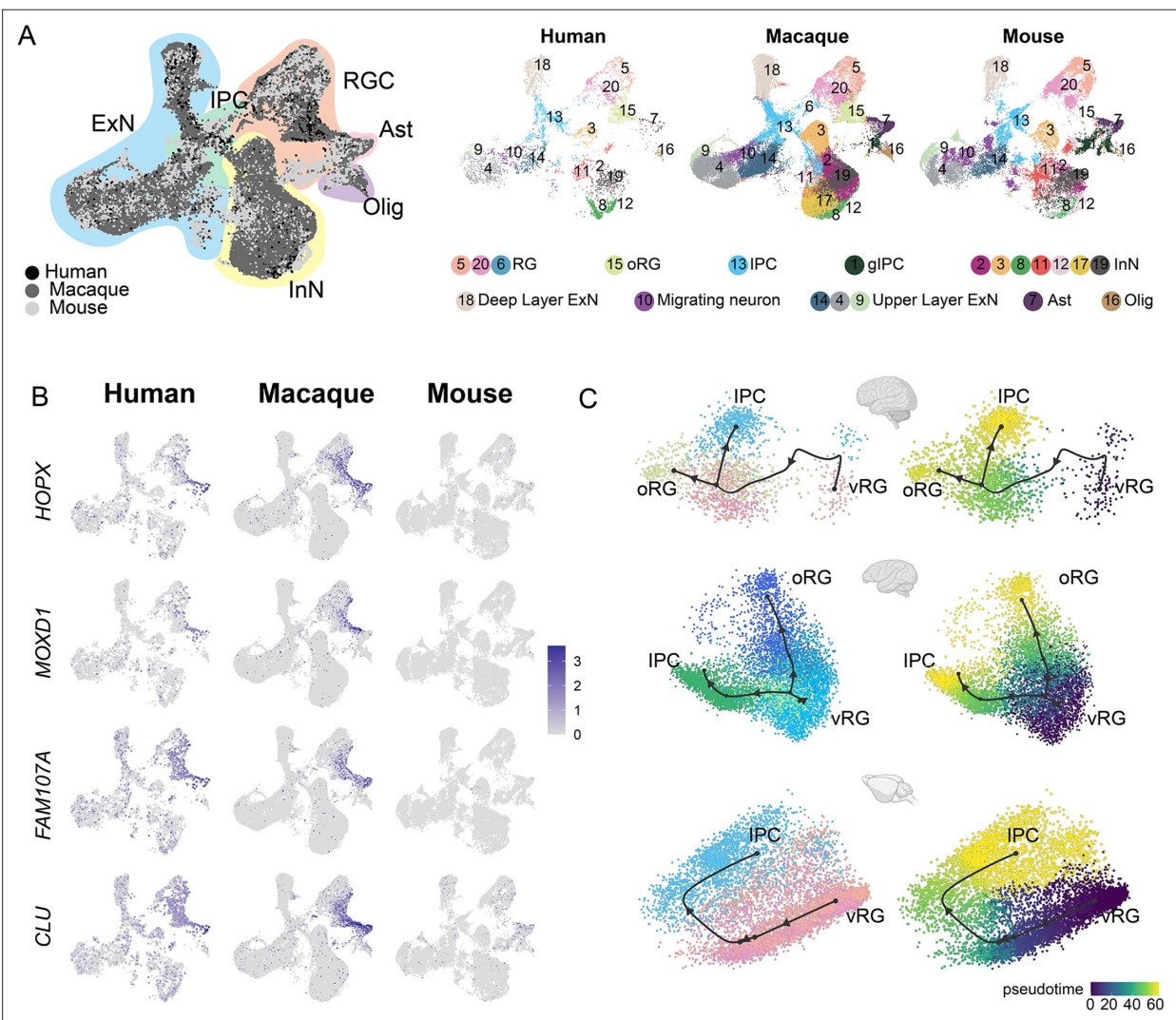

**Figure 5.** Integration of human, macaque, and mouse single-cell datasets reveals conserved and divergent progenitor cell types. (**A**) Left, uniform manifold approximation and projection (UMAP) plot of cross-species integrated single-cell transcriptome data with LIGER. Colors represent different major cell types (black: human dataset; dark gray: macaque dataset; lighter gray: mouse). Right, the UMAP plot of each dataset, colored according to the LIGER cluster. (**B**) The expressions of the classic outer radial glia (oRG) marker genes were plotted to UMAP visualization. Light gray, no expression; dark blue, relative expression. (**C**) Comparison of vRG→oRG and vRG→IPC developmental trajectories between human, macaque, and mouse.

The online version of this article includes the following figure supplement(s) for figure 5:

**Figure supplement 1.** Upper-layer and deep-layer excitatory neuron proportion analysis among species.

distinct, branch-specific sets of DEGs suggested a role for these cell-type-specific TFs and other genes in terminal neuron differentiation.

## Conserved and divergent features of human, macaque, and mouse neocortical progenitor cells

Next, we performed LIGER integration (*Welch et al., 2019*) to integrate the macaque single-cell dataset with published mouse (*Di Bella et al., 2021*) and human datasets, which were created by combing two published human scRNA-seq data of 7–21 postconceptional week (PCW) neocortex (*Zhong et al., 2018*; *Fan et al., 2020*), to conduct a cross-species comparison. Species-integrated UMAP showed that all major cell types were well integrated between the three species (*Figure 5A*).

It is clearly observed that similar cellular processes with comparable temporal progression sequentially generate deep-layer and upper-layer neurons in both primates and rodents (*Figure 5—figure supplement 1A to C*). However, in primates, the number and relative development period of upper neurons are more extended than that of rodents (*Figure 5—figure supplement 1D*). Previous study evidence had shown that extending neurogenesis by transplanting mouse embryos to a rat mother increases explicitly the number of upper-layer cortical neurons, with concomitant abundant neurogenic progenitors in the sub-VZ (*Stepien et al., 2020*). We thought this mechanism might also explain primates' much more expanded abundance of upper-layer neurons. Besides, another important evolutionarily divergent feature in cortical development between primates and rodents is the abundance of oRG cells in primates' OSVZ (human and macaque). Our cross-species analysis showed strong expression of the oRG marker genes, such as *HOPX*, *MOXD1*, *FAM107A,* and *CLU* in both human and macaque datasets (*Figure 5B*), while barely detectable expression in mouse dataset, which is consistent with previous reports (*Wang et al., 2011*).

Furthermore, we picked *HOPX*-positive and *EOMES*-positive cells (LIGER clusters 5, 20, 13, 15) from the human, macaque, and mouse datasets and performed developmental trajectories analysis with Slingshot (*Figure 5C*). Comparing vRG-to-IPC trajectory between human, macaque, and mouse, we found this biological process of vRG-to-IPC is remarkably conserved across species. However, the vRG-to-oRG trajectory is divergent between species because the oRG population was not identified in the mouse dataset. The latter process is almost invisible in mice but similar in humans and macaques.

Previous studies have shown that neural progenitors exhibit distinct temporal expression patterns during neurogenesis in mice. However, similar temporal profiling has yet to be thoroughly investigated in macaque neural stem cells. Based on its topological position and associated markers that suggest it functions as a developmental root, we examined the temporal expression profile of RG_C10 cells (*Figure 3A*). We first identified marker genes to distinguish RG_C10 cells in different prenatal stages (i.e. E40–E90) and used these markers to generate dynamic expression profiles. We then clustered these dynamically expressed macaque genes into five types (*Figure 6A*), consisting of Type 1 and 2 genes that decreased in expression during neurogenesis, Type 3, 4, and 5 genes that gradually ramped up in expression throughout neurogenesis. We then performed a similar profiling of dynamic gene expression using data obtained from mouse apical progenitors (*Ruan et al., 2021*) and categorized the DEGs into the same five categories. A total of 72 homologous TFs were shared between human, macaque, and mouse transcriptomes. Analysis of their temporal dynamics revealed that they exhibited similar temporal patterns between species such as *NEUROD1*, *NEUROG2*, *POU3F2*, *ETV1*, *ETV5,* and *FOS* (*Figure 6B* and *Figure 6—figure supplement 1*), which indicated that the majority of TFs had conserved temporal dynamics between species, and thus evolutionarily conserved patterns of regulation in neural stem cell differentiation.

To identify the master regulators related to cortical neurogenesis among macaque and mouse, we used the SCENIC workflow to analyze the gene regulatory networks of each TF (*Figure 6C to H*). Among all the regulatory pairs associated with each prenatal stage, we selected 10 TFs with the highest regulon specificity scores (*Figure 6C, E, and G*) and their top 5 target genes, as visualized by Cytoscape. Analysis of the regulatory activity of human, macaque, and mouse (*Di Bella et al., 2021*) prenatal neocortical neurogenesis indicated commonalities in the roles of classical developmental TFs such as *GATA1*, *SOX2*, *HMGN3*, *TCF7L1*, *ZFX*, *EMX2*, *SOX10*, *NEUROG1*, *NEUROD1,* and *POU3F1*. The top 10 TFs and their top 5 target genes of the human, macaque, and mouse vRG at each time point were identified by pySCENIC as an input to construct the transcriptional regulation network (*Figure 6D, F, and H*). Some conserved regulatory TFs present in more than one species are identified,

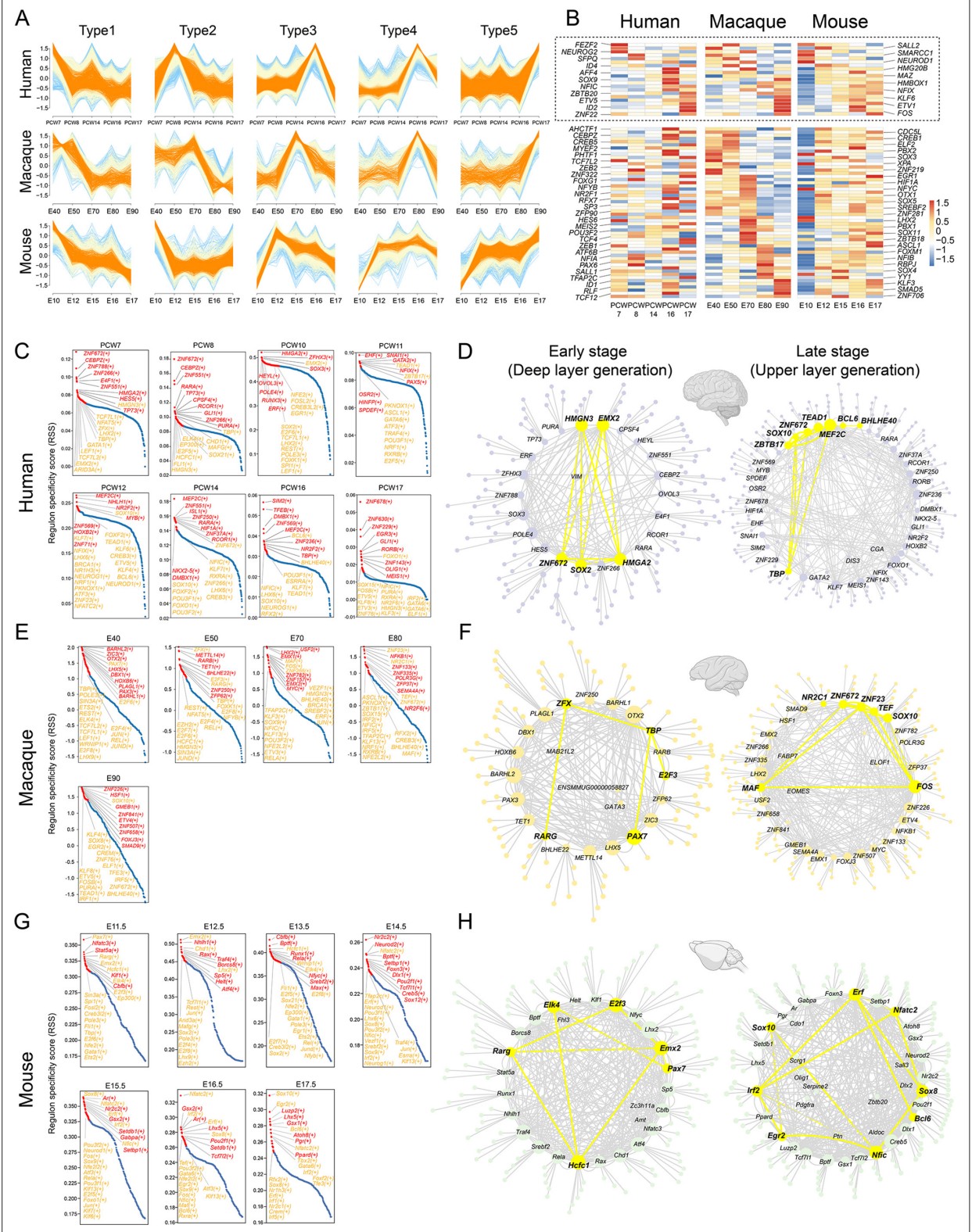

**Figure 6.** The patterns of transcriptional regulation comparative analysis responsible in ventricular radial glia (vRGs). (**A**) Normalized expressions of genes that show temporal dynamics in the vRGs of human, macaque, and mouse. (**B**) Temporal expression heatmap of homologous transcription factor (TF) genes among human, macaque, and mouse. (The TFs genes in the dashed boxes showed similar temporal expression patterns across species.) (**C**, **E**, and **G**) Regulon specificity score for each time point in human, macaque, and mouse vRG. Regulons with high scores in multiple species vRG cells are colored yellow. (**D**, **F**, and **H**) show a network generated with Cytoscape using the top 10 regulons in the human, macaque, and mouse vRG at each

*Figure 6 continued on next page*

*Figure 6 continued*

time point and their top 5 target genes identified by SCENIC as an input. The interactions between conserved TFs in more than one species are colored yellow.

The online version of this article includes the following figure supplement(s) for figure 6:

**Figure supplement 1.** Temporal expression pattern of RNA binding protein and transcription factor genes in human, macaque, and mouse ventricular radial glia (vRG).

such as *HMGN3*, *EMX2*, *SOX2*, and *HMGA2*, at the deep-layer generation stage and *SOX10*, *ZNF672*, and *ZNF672* at the upper-layer generation stage.

## Discussion

The cerebral cortex region of the brain is responsible for extraordinary cognitive capacities, such as abstract thinking and language. The parietal lobe is the center of the somatic senses and is significant for interpreting words as well as language understanding and processing. In this study, the parietal lobe area was selected mainly because of the convenience of sampling the dorsal neocortex. The neocortex of NHP rhesus macaque resembles that of humans in many aspects. Thus, a comprehensive investigation of macaque neurogenesis can improve our understanding of neocortical development and evolution. For this purpose, we performed scRNA-seq in prenatal neocortical tissues of macaque. Our data support previous omics studies of prenatal macaque brain development (*Bakken et al., 2016*; *Zhu et al., 2018*; *Luo et al., 2021*) and recently published human neocortex development studies (*Bhaduri et al., 2021*). We constructed a single-cell resolution transcriptomic atlas of the developing macaque neocortex, which we then used to identify dynamically expressed genes that likely contribute to the maturation of distinct neuron types, temporal expression patterns of neural stem cells, and the generation of oRGs.

The existence of oRGs has long been demonstrated in humans and primates through live imaging and immunostaining (*Hansen et al., 2010*). ORGs share similarities in their patterns of gene expression with vRGs (e.g. *SOX2*, *PAX6*, and *NESTIN*) but also specifically express several genes (such as *MOXD1*, *HOPX*, and *FAM107A*) (*Pollen et al., 2015*; *Nowakowski et al., 2016*). Recently, the molecular mechanisms associated with the generation and amplification of oRGs were shown to include signaling pathways such as the *FGF-MAPK* cascade, *SHH*, *PTEN/AKT*, and *PDGF* pathways, as well as proteins such as *INSM*, *GPSM2*, *ASPM*, *TRNP1*, *ARHGAP11B*, *PAX6*, and *HIF1α*. A number of these proteins were validated by genetic manipulation of their activities or expression in mouse, ferret, and marmoset (; *Kalebic et al., 2018*; *Xing et al., 2021*; *Heide et al., 2020*). The dynamically expressed genes and pseudotime trajectories from vRGs to oRGs presented in this work are in agreement with these previous reports. In addition to known regulators, we detected differential expression of transporters and ion channel regulators in oRGs, such as *ATP1A2*, *ATP1B2*, and *SCN4B*. Indeed, hyperpolarization in mouse apical progenitors has been shown to promote the generation of IPCs and indirect neurogenesis (*Vitali et al., 2018*), although it remains unclear if similar mechanisms contribute to the development of the primate neocortex.

The sequential generation of cortical neurons has long been observed in both rodents and primates, and many TFs are known to drive post-mitotic specification of neuronal progeny, mainly in rodents. Here, we found genes conserved across species, such as *FEZF2* and *TBR1*, involved in specifying deep-layer neurons in the macaque neocortex. The Slingshot pseudotime analysis, which reflected the expression pattern of genes during appropriate lineage trajectories, suggests that some genes are lineage-restricted or even involved in lineage fate determination. For example, *FEZF2*, *PPP1R1B* (also known as *DARPP32*), *SOX5*, *LMO3*, and *CELF4* may be involved in the fate specialization of deep neurons; in contrast, *TMOD1*, *PID1*, *LINGO1*, *CRYM*, and *MEF2C* for upper-layer neurons. Notably, *FEZF2* (*Molyneaux et al., 2005*) and *SOX5 Leone et al., 2008* have been confirmed associated with deep-layer neuron specification in mice, while others, such as *TMOD1*, *PID1*, and *LINGO1*, have not been reported before, and their potential functions in the fate specification of cortical projection neurons need to be further explored. To date, only a very limited set of genes have been shown to function in specifying neuronal progeny in progenitor cells, which suggests that regulation in progenitors is more complex than in neurons, where a single TF can trigger a transcriptional cascade to promote the maturation of neurons. In the current study, we found two types of genes (Dynamic type

1 and 5), which have gradually altered expression patterns along with the neurogenic stage, indicating a transition in cell state in both vRGs and oRGs. In addition to these gradual transition genes, we also identified sets of genes (Dynamic types 2–4) with sharp changes in expression during neurogenesis, which likely function in specifying different neuronal subtypes. These findings led us to postulate that a combination of biological processes and pathways change over time to coordinate gradual transitions from neural stem cells to daughter neurons.

This study also faces limitations. During the macaque prenatal neurogenesis, dorsal RG cells generate ENs by direct and indirect neurogenesis. In general, the number of progenitor cells gradually decreases throughout development, with progenitor cells generating deep neurons in the early stage of development and upper ENs in the later stages. We observed the disappearance of two subclusters of ENs (EN5 and EN10) from later-stage samples (*Figure 2—figure supplement 1*). Besides, the number of neurons at the E90 time point was relatively small. This disappearance could also be explained by the development of axons and dendrites, which have higher morphological complexity and greater vulnerability, because mature neurons at E90 with abundant axons and processes were hard to settle into micropores of the BD method for single-cell capture. Due to the fixed size of the BD Rhapsody microwells, this sing-cell capture method might be less efficient in capturing mature ENs but has a good capture effect on newborn neurons at each sampling time point. In conclusion, based on the BD cell capture method feature, the immature neurons at each point are more easily captured than mature neurons in our study, so the generation of ENs at different developmental time points can be well observed, as shown in *Figure 2*, which aligns with our research purpose. Nonetheless, this transcriptomic atlas uncovers the molecular signatures of the primary cell types and temporal shifts in gene expression of differentiating progenitor cells during neocortex layer formation, thus providing a global perspective into neocortical neurogenesis in the macaque.

## Materials and methods
### Animal
Animals and frozen tissue samples from prenatal rhesus macaque (*Macaca mulatta*) were provided by the National Medical Primate Center, Institute of Medical Biology, Chinese Academy of Medical Sciences. Timed pregnancy-derived biological replicate specimens were profiled at each prenatal developmental stage (E40, E50, E70, E80, and E90) (*Bakken et al., 2016*). These time points were selected to coincide with peak periods of neurogenesis for the different layers of the neocortex based on previous studies (*Bakken et al., 2016* ; *Lein et al., 2017*). All animal procedures followed international standards and were approved in advance by the Ethics Committee on Laboratory Animals at IMBCAMS (Institute of Medical Biology, Chinese Academy of Medical Sciences).

### Fetal brain sample details
We collected eight pregnancy-derived fetal brains of rhesus macaque (*M. mulatta*) at five prenatal developmental stages (E40, E50, E70, E80, E90) and dissected the parietal lobe cortex. Because of the different development times of rhesus monkeys, prenatal cortex size and morphology are different. To ensure that the anatomical sites of each sample are roughly the same, we use the lateral groove as a reference to collect the parietal lobe for single-cell sequencing (as indicated by bright yellow in *Figure 1—figure supplement 1A*) and do not make a clear distinction between the different regional parts including primary somatosensory cortex and association cortices in the process of sampling.

### Cell preparation
We followed the BD Cell Preparation Guide to wash, count, and concentrate cells in preparation for use in BD Rhapsody System Whole Transcriptome Analysis.

### scRNA-seq data processing
Single-cell capture was achieved by random distribution of a single-cell suspension across >200,000 microwells through a limited dilution approach. Beads with oligonucleotide barcodes were added to saturation so that a bead was paired with a cell in a microwell. The cell-lysis buffer was added so that poly-adenylated RNA molecules hybridized with the beads. Beads were collected into a single tube for reverse transcription. Upon cDNA synthesis, each cDNA molecule was tagged on the 5′ end (the

3′ end of a mRNA transcript) with a molecular index and cell label indicating its cell of origin. Whole-transcriptome libraries were prepared using the BD Resolve single-cell whole-transcriptome amplification workflow. In brief, the second strand of cDNA was synthesized, followed by ligation of the adaptor for universal amplification. Eighteen cycles of PCR were used to amplify the adaptor-ligated cDNA products. Sequencing libraries were prepared using random priming PCR of the whole-transcriptome amplification products to enrich the 3′ ends of the transcripts linked with the cell label and molecular indices. Sequencing was performed with Illumina NovaSeq 6000 according to the manufacturer's instructions. The filtered reads were aligned to the macaque reference genome file with Bowtie2 (v2.2.9). Macaque reference genome was downloaded from Ensemble database (Macaque reference genome). We analyzed the sequencing data following BD official pipeline and obtained the cell-gene expression matrix file.

## QC and data analysis

To mitigate the effect of over-estimation of molecules from PCR and sequencing errors, we used the Unique Molecular Identifier adjustment algorithms, recursive substitution error correction, and distribution-based error correction, which were contained in the BD Rhapsody pipeline. Quality control was applied based on the detected gene number and the percentage of counts originating from mitochondrial RNA, ribosomal RNA, and hemoglobin gene per cell. Then, cells were filtered to retain only higher quality (with less than 7.5% mitochondrial gene counts, less than 5.5% ribosomal gene counts, and detected genes above 400 and less than 6000). Additionally, we identified cell doublets using Scrublet v0.142 with default parameters and removed doublets before data analysis. After quality control, a total of 53,259 cells remained for subsequent analysis.

To integrate cells into a shared space from different datasets for unsupervised clustering, we used SCTransform workflow in R packages Seurat (*Satija et al., 2015*) v4.0.3 to normalize the scRNA-seq data from different samples. We identified the variable features of each donor sequencing data using the 'FindVariableFeatures' function. A consensus list of 2000 variable genes was then formed by detecting the most significant recovery rates genes across samples. Mitochondrial and ribosomal genes were not included in the variable gene list. Next, we used the SCTransform workflow in Seurat to normalize the single-cell sequence data from different samples. During normalization, we also removed confounding sources of variation, including mitochondrial mapping and ribosomal mapping percentages. Standard RPCA workflow was used to perform the integration.

With a 'resolution' of 0.5 upon running 'FindClusters', we distinguished major cell types of nerve cells and non-nerve cells in the UMAP according to known markers, including RG cell, ORG cell, IPC, EN, IN, astrocyte, oligodendrocyte, Cajal-Retzius cell, microglia, endothelial cell, meningeal cells, and blood cells, which were distinguished on the first level. By following a similar pipeline, subclusters were identified. We finalized the resolution parameter on the FindClusters function once the cluster numbers did not increase when we increased the resolution. Then, we checked the DEGs between each of the clusters using the 'FindAllMarkers' and 'FindMarkers' functions with logfc.threshold=0.25.

## Construction of single-cell developmental trajectory

Single-cell developmental lineage trajectories construction and discovery of trajectory transitions were performed using Slingshot (v2.2.0) (*Street et al., 2018*), from the Dyno (v0.1.2) (*Saelens et al., 2019*) platform, with PCA dimensionality reduction plot results. The direction of the developmental trajectory was adjusted by reference to the verified relevant studies.

## Cross-species transcriptome data integration and analysis

Mouse datasets were downloaded from the Gene Expression Omnibus (GEO SuperSeries GSE153164) and at the Single Cell Portal: here. Human datasets were download from the GEO under the accession number GSE104276 and GSE104276. We created a human database by combining the two published human prenatal cortical development datasets of 7–21 PCW into one containing cell numbers comparable to our macaque and published mouse dataset using the single-cell transcriptome analysis pipeline of R packages Seurat. We used the biomaRt package to convert the gene symbols in the mouse and macaque expression matrices into their human homologs. To perform cross-species analysis, we used the LIGER (*Telley et al., 2019*) method to integrate the macaque single-cell dataset with the mouse and human datasets. We ran the

'optimizeALS' function in rliger to perform integrative non-negative matrix factorization on the scaled species-integrated dataset (k=20, lambda = 5, max.iters=30). Species-integrated UMAP showed that all major cell types were well integrated between the three species (*Figure 5A*). We used the 'calcAlignment' function in rliger to quantify how well-aligned datasets are and got the metric of 0.891231545756746, which was very close to '1'. This suggested LIGER well-integrated cross-species datasets.

## TF regulatory network analysis

We inferred regulon activities for vRG cells of human, macaque, and mouse using pySCENIC (*Verfaillie et al., 2015*) workflow (https://pyscenic.readthedocs.io/en/latest/tutorial.html) with default parameters. Since there is no transcriptional regulator database for macaques, macaques' genome is similar to the human genome. When we analyzed the macaque dataset, we used the human transcriptional regulator database for pySCENIC analysis. GRNboost2 in SCENIC was used to infer gene regulatory networks based on co-expression patterns. RcisTarget (*Verfaillie et al., 2015*) was used to analyze the TF-motif enrichment and direct targets among different species databases. The regulon activity for each developmental time point was identified using AUCell, and the top enriched activated TFs were ranked by -log10(p_value). Lastly, we evaluated the interaction networks analysis of TFs and their target genes by Cytoscape (*Shannon et al., 2003*) (v3.8.1).

## Mfuzz clustering

Firstly, we picked up vRG cells in each species dataset and screened the DEGs between adjacent development time points using the 'FindMarkers' function (with min.pct=0.25, logfc.threshold=0.25). After separate normalization of the DEG expression matrix from different species datasets, we use the 'standardise' function of Mfuzz to perform standardization. The DEGs of vRG in each species were grouped into five different clusters using the Mfuzz package in R with fuzzy c-means algorithm (*Kumar and Futschik, 2007*).

## TF and RNA binding protein analysis

Mouse, macaque, and human TF gene lists were downloaded from AnimalTFDB (http://bioinfo.life.hust.edu.cn/AnimalTFDB/). Mouse, macaque, and human RNA binding protein (RBP) gene lists were downloaded from EuRBPDB (http://EuRBPDB.syshospital.org). Heatmaps of the TFs and RBPs expression profiles were generated with normalized and standardized DEG expression matrices from Mfuzz.

## Acknowledgements

We thank members of the Peng and Chen laboratories for helpful discussions. This work was supported by CAMS Innovation Fund for Medical Sciences (CIFMS, 2021- I2M- 1- 024, 2021- I2M- 1- 019), National Science and Technology Innovation 2030 Major Program 2021ZD0200902 and Overseas Expertise Introduction Center for Discipline Innovation ("111Center") BP0820029.

.

## Additional information

### Funding

| Funder | Grant reference number | Author |
| --- | --- | --- |
| China Academy of Medical Science | 2021-I2M-1-024 | Xiaozhong Peng |
| China Academy of Medical Science | 2021-I2M-1-019 | Pengcheng Shu |
| National Science and Technology Major Project | 2021ZD0200902 | Xiaozhong Peng |

| Funder | Grant reference number | Author |
|---|---|---|
| Overseas Expertise Introduction Center for Discipline Innovation ("111Center") | BP0820029 | Xiaozhong Peng |

The funders had no role in study design, data collection and interpretation, or the decision to submit the work for publication.

## Author contributions

Longjiang Xu, Data curation, Validation, Investigation, Visualization, Writing – original draft, Writing – review and editing; Zan Yuan, Data curation, Software, Visualization, Writing – original draft; Jiafeng Zhou, Visualization, Writing – original draft; Yuan Zhao, Conceptualization, Resources; Wei Liu, Conceptualization, Resources, Investigation; Shuaiyao Lu, Resources, Supervision; Zhanlong He, Resources; Boqin Qiang, Conceptualization, Supervision; Pengcheng Shu, Conceptualization, Project administration, Writing – review and editing; Yang Chen, Conceptualization, Supervision, Methodology, Writing – review and editing; Xiaozhong Peng, Conceptualization, Resources, Supervision, Funding acquisition, Methodology, Project administration, Writing – review and editing

## Author ORCIDs

Longjiang Xu ⓘ http://orcid.org/0000-0002-3534-6840
Pengcheng Shu ⓘ http://orcid.org/0000-0002-3091-2559
Xiaozhong Peng ⓘ http://orcid.org/0000-0002-9592-9554

Reviewer #1 (Public Review): https://doi.org/10.7554/eLife.90325.3.sa1
Reviewer #2 (Public Review): https://doi.org/10.7554/eLife.90325.3.sa2
Reviewer #3 (Public Review): https://doi.org/10.7554/eLife.90325.3.sa3
Author Response https://doi.org/10.7554/eLife.90325.3.sa4

# Additional files

## Supplementary files

• Supplementary file 1. Marker list of 28 cell clusters.

• Supplementary file 2. Gene expression importance across the ventricular radial glia (vRG) to outer radial glia (oRG) (C10–C14–C12) lineage trajectory related to *Figure 3E* (vRG: milestone2; oRG: milestone4).

• Supplementary file 3. Gene expression importance across the radial glia (RG) to deep-layer neurons and upper-layer lineage trajectory related to *Figure 4D*.

• Supplementary file 4. Gene expression importance across the ventricular radial glia (vRG) to intermediate progenitor cell (IPC) (C10–C22–C8) trajectory related to *Figure 3—figure supplement 1B*.

• Supplementary file 5. Gene expression importance across the glia intermediate precursor cell (gIPC) differentiation into astrocytes and oligodendrocytes lineage trajectory related to *Figure 3—figure supplement 2E*.

• Supplementary file 6. The original data of normalized gene expression matrix in human, macaque, and mouse ventricular radial glia (vRG) related to *Figure 6A*.

• Supplementary file 7. Regulon specificity score for each time point in human, macaque, and mouse ventricular radial glia (vRG) related to *Figure 6C to H*.

• MDAR checklist

## Data availability

The raw sequence data reported in this paper have been deposited in the Genome Sequence Archive in National Genomics Data Center, China National Center for Bioinformation / Beijing Institute of Genomics, Chinese Academy of Sciences (PRJCA008997) that are publicly accessible at https://ngdc.cncb.ac.cn/bioproject/browse/PRJCA008997. Codes used to analyze results in this paper are available on GitHub at https://github.com/cheneylemon/developing-macaque-neocortex, (copy archived at *Xu, 2023*).

The following dataset was generated:

| Author(s) | Year | Dataset title | Dataset URL | Database and Identifier |
|-----------|------|---------------|-------------|------------------------|
| Xu L, Peng X | 2024 | Single-cell transcriptional atlas of the macaque across the embryonic cortical layers development | https://ngdc.cncb. ac.cn/bioproject/ browse/PRJCA008997 | China National Center for Bioinformation, PRJCA008997 |

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
