## [Editor Report · eLife assessment]

This study presents a **useful** resource for the gene expression profiles of different cell types in the parietal lobe of the cerebral cortex of prenatal macaques. The evidence supporting the claims of the authors is **solid**, and revision has clarified some of the cell isolation and cell classification issues flagged by reviewers. This dataset will be of interest to developmental neurobiologists and could potentially be used for future comparative studies on early brain development.

---

## [Referee Report · Reviewer #1 (Public Review)]

In the article "Temporal transcriptomic dynamics in developing macaque neocortex", Xu et al. analyze the cellular composition and transcriptomic profiles of the developing macaque parietal cortex using single-cell RNA sequencing. The authors profiled eight prenatal rhesus macaque brains at five timepoints (E40, E50, E70, E80, and E90) and obtained a total of around 53,000 high-quality cells for downstream analysis. The dataset provides a high-resolution view into the developmental processes of early and mid-fetal macaque cortical development and will potentially be a valuable resource for future comparative studies of primate neurogenesis and neural stem cell fate specification. Their analysis of this dataset focused on the temporal gene expression profiles of outer and ventricular radial glia and utilized pesudotime trajectory analysis to characterize the genes associated with radial glial and neuronal differentiation. The rhesus macaque dataset presented in this study was then integrated with prenatal mouse and human scRNA-seq datasets to probe species differences in ventricular radial glia to intermediate progenitor cell trajectories. Additionally, the expression profile of macaque radial glia across time was compared to those of mouse apical progenitors to identify conserved and divergent expression patterns of transcription factors.

The main findings of this paper corroborate many previously reported and fundamental features of primate neurogenesis: deep layer neurons are generated before upper layer excitatory neurons, the expansion of outer radial glia in the primate lineage, conserved molecular markers of outer radial glia, and the early specification of progenitors. Furthermore, the authors show some interesting divergent features of macaque radial glial gene regulatory networks as compared to mouse. Overall, despite some uncertainties surrounding the clustering and annotations of certain cell types, the manuscript provides a valuable scRNA-seq dataset of early prenatal rhesus macaque brain development. The dynamic expression patterns and trajectory analysis of ventricular and outer radial glia provide valuable data and lists of differentially expressed genes (some consistent with previous studies, others reported for the first time here) for future studies.

---

## [Referee Report · Reviewer #2 (Public Review)]

Summary:

This manuscript by Xu et al., is an interesting study aiming to identify novel features of macaque cortical development. This study serves as a valuable atlas of single cell data during macaque neurogenesis, which extends the developmental stages previously explored. Overall, the authors have achieved their aim of collecting a comprehensive dataset of macaque cortical neurogenesis and have identified a few unknown features of macaque development.

Strengths:

The authors have accumulated a robust dataset of developmental time points and have applied a variety of informatic approaches to interrogate this dataset. One interesting finding in this study is the expression of previously unknown receptors on macaque oRG cells. Another novel aspect of this paper is the temporal dissection of neocortical development across species. The identification that the regulome looks quite different, despite similar expression of transcription factors in discrete cell types, is intriguing.

---

## [Referee Report · Reviewer #3 (Public Review)]

Summary:

The study adds to the existing data that have established that cortical development in rhesus macaque is known to recapitulate multiple facets cortical development in humans. The authors generate and analyze single cell transcriptomic data from the timecourse of embryonic neurogenesis.

Strengths:

Studies of primate developmental biology are hindered by the limited availability and limit replication. In this regard, a new dataset is useful.

The study analyzes parietal cortex, while previous studies focused on frontal and motor cortex. This may be the first analysis of macaque parietal cortex and, as such, may provide important insights into arealization, which the authors have not addressed

---

## [Author Response]

Thanks to all the reviewers for their insightful and constructive comments, which are very helpful in improving the manuscript. We are encouraged by the many positive comments regarding the significance of our findings and the value of our data. Regarding the reviews’ concern on cell classification, we used several additional marker genes to explain the identification of cell clusters and subclusters. We have further analyzed and rewrote part of the text to address the concerns raised. Here is a point-by-point response to the reviewers’ comments and concerns. Figures R1-R9 were provided only for additional information for reviewers and were not included in the revisedmanuscript.

**Reviewer #1 (Public Review):**
In the article "Temporal transcriptomic dynamics in developing macaque neocortex", Xu et al. analyze the cellular composition and transcriptomic profiles of the developing macaque parietal cortex using single-cell RNA sequencing. The authors profiled eight prenatal rhesus macaque brains at five timepoints (E40, E50, E70, E80, and E90) and obtained a total of around 53,000 high-quality cells for downstream analysis. The dataset provides a high-resolution view into the developmental processes of early and mid-fetal macaque cortical development and will potentially be a valuable resource for future comparative studies of primate neurogenesis and neural stem cell fate specification. Their analysis of this dataset focused on the temporal gene expression profiles of outer and ventricular radial glia and utilized pesudotime trajectory analysis to characterize the genes associated with radial glial and neuronal differentiation. The rhesus macaque dataset presented in this study was then integrated with prenatal mouse and human scRNA-seq datasets to probe species differences in ventricular radial glia to intermediate progenitor cell trajectories. Additionally, the expression profile of macaque radial glia across time was compared to those of mouse apical progenitors to identify conserved and divergent expression patterns of transcription factors.The main findings of this paper corroborate many previously reported and fundamental features of primate neurogenesis: deep layer neurons are generated before upper layer excitatory neurons, the expansion of outer radial glia in the primate lineage, conserved molecular markers of outer radial glia, and the early specification of progenitors. Furthermore, the authors show some interesting divergent features of macaque radial glial gene regulatory networks as compared to mouse. Overall, despite some uncertainties surrounding the clustering and annotations of certain cell types, the manuscript provides a valuable scRNA-seq dataset of early prenatal rhesus macaque brain development. The dynamic expression patterns and trajectory analysis of ventricular and outer radial glia provide valuable data and lists of differentially expressed genes (some consistent with previous studies, others reported for the first time here) for future studies.The major weaknesses of this study are the inconsistent dissection of the targeted brain region and the loss of more mature excitatory neurons in samples from later developmental timepoint due to the use of single-cell RNA-seq. The authors mention that they could observe ventral progenitors and even midbrain neurons in their analyses. Ventral progenitors should not be present if the authors had properly dissected the parietal cortex. The fact that they obtained even midbrain cells point to an inadequate dissection or poor cell classification. If this is the result of poor classification, it could be easily fixed by using more markers with higher specificity. However, if it is the result of a poor dissection, some of the cells in other clusters could potentially be from midbrain as well. The loss of more mature excitatory neurons is also problematic because on top of hindering the analysis of these neurons in later developmental periods, it also affects the cell proportions the authors use to support some of their claims. The study could also benefit from the validation of some of the genes the authors uncovered to be specifically expressed in different populations of radial glia.

We thank the Reviewer’s comments and apologize for the shortcomings of tissue dissection and cell capture.

We used more marker genes for major cell classification, such as SHOX2, IGFBP5, TAC1, PNYN, FLT1, and CYP1B, in new Figure 1D, to improve the cell type annotation results. We improved the cell type annotation results by fixing cluster 20 from C20 as Ventral LGE-derived interneuron precursor and cluster by the expression of IGFBP5, TAC1, and PDYN; fixing cluster 23 from meningeal cells to thalamus cells by the expression of ZIC2, ZIC4, and SHOX2. These cell types were excluded in the follow-up analysis. Due to EN8 being previously incorrectly defined as midbrain neurons, it resulted in a misunderstanding of the dissection result as a poor dissection. After carefully reviewing the data analysis process, we determined that EN8 was a small group of cells in cluster 23 mistakenly selected during excitatory neuron analysis, as shown in Figure R5(A), which was corrected after revision. In the revised manuscript, we deleted the previous EN8 subcluster and renumbered the rest of the excitatory neuron subclusters in the new Figure 2.

In addition, we also improved the description of sample collection as follows: “We collected eight pregnancy-derived fetal brains of rhesus macaque (*Macaca mulatta*) at five prenatal developmental stages (E40, E50, E70, E80, E90) and dissected the parietal lobe cortex. Because of the different development times of rhesus monkeys, prenatal cortex size and morphology are different. To ensure that the anatomical sites of each sample are roughly the same, we use the lateral groove as a reference to collect the parietal lobe for single-cell sequencing (as indicated by bright yellow in Figure S1A) and do not make a clear distinction between the different regional parts including primary somatosensory cortex and association cortices in the process of sampling”. As shown in Figure S1A, due to the small volume of the cerebral cortex at early time points, especially in E40, a small number of cells beyond the dorsal parietal lobe, including the ventral cortex cells and thalamus cells, were collected during the sampling process with the brain stereotaxic instrument.

In this study, the BD method was used to capture single cells. Due to the fixed size of the micropores, this method might be less efficient in capturing mature excitatory neurons. However, it has a good capture effect on newborn neurons at each sampling time point so that the generation of excitatory neurons at different developmental time points can be well observed, as shown in Figure 2, which aligns with our research purpose.

To verify the reliability of our cell annotation results, we compared the similarity of cell-type association between our study and recently published research(Micali N, Ma S, Li M, et al. Science. doi:10.1126/science.adf3786.PMID: 37824652), using the scmap package to project major cell types in our macaque development scRNA-seq dataset to GSE226451. The river plot in Author response image 1 illustrates the broadly similar relationships of cell type classification between the two datasets.

**Author response image 1. sa4fig1:** Riverplot illustrates relationships between datasets in this study and recently published developing macaque telencephalon datasets major cell type annotation.

Furthermore, bioinformatics analysis is used for the validation of genes specifically expressed in outer radial glia. We verified terminal oRG differentiation genes in the recently published macaque telencephalic development dataset(Micali N, Ma S, Li M, et al. Science. doi:10.1126/science.adf3786.PMID: 37824652) (GEO accession: GSE226451). The results of Author response image 2 show that the gene expression showed states/stages. Most of the oRG terminal differentiation markers genes identified in our study were also expressed in the oRG cells of the GSE226451 dataset. In particular, the two datasets were consistent in the expression of ion channel genes ATP1A2, ATP1A2, and SCN4B.

**Author response image 2. sa4fig2:** Heatmap shows the relative expression of genes displaying significant changes along the pseudotime axis of vRG to oRG from the dataset of Nicola Micali et al. 2023(GEO: GSE226451). The columns represent the cells being ordered along the pseudotime axis.

**Reviewer #2 (Public Review):**
Summary:This manuscript by Xu et al., is an interesting study aiming to identify novel features of macaque cortical development. This study serves as a valuable atlas of single cell data during macaque neurogenesis, which extends the developmental stages previously explored. Overall, the authors have achieved their aim of collecting a comprehensive dataset of macaque cortical neurogenesis and have identified a few unknown features of macaque development.Strengths:The authors have accumulated a robust dataset of developmental time points and have applied a variety of informatic approaches to interrogate this dataset. One interesting finding in this study is the expression of previously unknown receptors on macaque oRG cells. Another novel aspect of this paper is the temporal dissection of neocortical development across species. The identification that the regulome looks quite different, despite similar expression of transcription factors in discrete cell types, is intriguing.Weaknesses:Due to the focus on demonstrating the robustness of the dataset, the novel findings in this manuscript are underdeveloped. There is also a lack of experimental validation. This is a particular weakness for newly identified features (like receptors in oRG cells). It's important to show expression in relevant cell types and, if possible, perform functional perturbations on these cell types. The presentation of the data highlighting novel findings could also be clarified at higher resolution, and dissected through additional informatic analyses. Additionally, the presentation of ideas and goals of this manuscript should be further clarified. A major gap in the study rationale and results is that the data was collected exclusively in the parietal lobe, yet the rationale and interpretation of what this data indicates about this specific cortical area was not discussed. Last, a few textual errors about neural development are also present and need to be corrected.

We thank you for your comments and suggestions concerning our manuscript. The comments and suggestions are all valuable and helpful for revising and improving our paper and the essential guiding significance to our research. We have studied the comments carefully and made corrections, which we hope to meet with approval. We have endeavored to address the multiple points raised by the referee.

To support the reliability of our data and newly identified features, we verified terminal oRG differentiation genes in the recently published macaque telencephalic development dataset(Micali N, Ma S, Li M, et al. Science. doi:10.1126/science.adf3786.PMID: 37824652) (GEO accession: GSE226451). The results of Figure R2 show that the gene expression showed states/stages. Most of the oRG terminal differentiation markers genes identified in our study were also expressed in the oRG cells of the GSE226451 dataset. In particular, the two datasets were consistent in the expression of ion channel genes ATP1A2, ATP1A2, and SCN4B.

Our research results mainly explore the conserved features of neocortex development across species. By comparing evolution, we found the types of neural stem cells in the intermediate state, their generative trajectories, and gene expression dynamics accompanying cell trajectories. We further explored the stages of transcriptional dynamics during vRG generating oRG. More analysis was performed through transcriptional factor regulatory network analysis. We performed the TFs regulation network analysis of human vRG with pyscenic workflow. The top transcription factors of every time point in human vRG were calculated, and we used the top 10 TFs and their top 5 target genes to perform interaction analysis and generate the regulation network of human vRG in revised figure 6. In comparison of the pyscenic results of mouse, macaque and human vRG, it was obvious that the regulatory networks were not evolutionarily conservative. Compared with macaque, the regulatory network of transcription factors and target genes in humans is more complex. Some conserved regulatory relationships present in more than one species are identified, such as HMGN3, EMX2, SOX2, and HMGA2 network at an early stage when deep lager generation and SOX10, ZNF672, ZNF672 network at a late stage when upper-layer generation.

Although the parietal lobe is the center of the somatic senses and is significant for interpreting words as well as language understanding and processing. In this study, the parietal lobe area was selected mainly because of the convenience of sampling the dorsal neocortex. As we described in the Materials and Methods section as follows: “Because of the different development times of rhesus monkeys, prenatal cortex size and morphology are different. To ensure that the anatomical sites of each sample are roughly the same, we use the lateral groove as a reference to collect the parietal lobe for single-cell sequencing (as indicated by bright yellow in Figure S1A) and do not make a clear distinction between the different regional parts including primary somatosensory cortex and association cortices in the process of sampling”.

Thanks for carefully pointing out our manuscript's textual errors about neural development.We have corrected them which were descripted in the following response.

**Reviewer #3 (Public Review):**
Summary: The study adds to the existing data that have established that cortical development in rhesus macaque is known to recapitulate multiple facets cortical development in humans. The authors generate and analyze single cell transcriptomic data from the timecourse of embryonic neurogenesis.Strengths:Studies of primate developmental biology are hindered by the limited availability and limit replication. In this regard, a new dataset is useful.The study analyzes parietal cortex, while previous studies focused on frontal and motor cortex. This may be the first analysis of macaque parietal cortex and, as such, may provide important insights into arealization, which the authors have not addressed.Weaknesses:The number of cells in the analysis is lower than recent published studies which may limit cell representation and potentially the discovery of subtle changes.The macaque parietal cortex data is compared to human and mouse pre-frontal cortex. See data from PMCID: PMC8494648 that provides a better comparison.A deeper assessment of these data in the context of existing studies would help others appreciate the significance of the work.

We thank the reviewer for these suggestions and constructive comments. We agree with the reviewer that the cell number in our study is lower than in recently published studies. The scRNA sequencing in this study was completed between 2018 and 2019, the early stages of the single-cell sequencing technology application. Besides, we have been unable to get extra macaque embryos to enlarge the sample numbers recently since rhesus monkey samples are scarce. Therefore, the number of cells in our study is relatively small compared to recently published single-cell studies.

The dataset suggested by the reviewers is extremely valuable, and we tried to perform analysis as the reviewer suggested to explore temporal expression patterns in different species of parietal cortex. The dataset from PMCID: PMC8494648 provides the developing human brain across regions from gestation week (GW)14 to gestation week (GW)25. Since this data set only covers the middle and late stages of embryonic neurogenesis, it did not fully match the developmental time points of our study for integration analysis. However, we quoted the results of this study in the discussion section.

The human regulation analysis with pyscenic workflow was added into new figure 6 for the comparison of different species vRG regulatory network. Compared with macaque, the regulatory network of transcription factors and target genes in humans is more complex. Some conserved regulatory relationships present in more than one species are identified, such as HMGN3, EMX2, SOX2, and HMGA2 network at an early stage when deep lager generation and SOX10, ZNF672, ZNF672 network at a late stage when upper-layer generation.

Besides, we performed additional integration analysis of our dataset with the recently published macaque neocortex development datase (GEO accession: GSE226451) to verify the reliability of our cell annotation results and terminal oRG differentiation genes. The river plot in Figure R1 illustrates the broadly similar relationships of cell type classification between the two datasets. The result in Figure R2 showed that most of the oRG terminal differentiation markers genes identified in our study were also expressed in the oRG cells of the GSE226451 dataset. In particular, the two datasets were consistent in the expression of ion channel genes ATP1A2, ATP1A2, and SCN4B.

**Reviewer #1 (Recommendations For The Authors):**
1. Throughout the manuscript, the term "embryonic" or "embryogenesis" is used in reference to all timepoints (E40-E90) in this study. The embryonic period is a morphologically and anatomically defined developmental period that ends ~E48-E50 in rhesus macaque. Prenatal or developing is a more accurate term when discussing all timepoints of this study.

We thank the reviewer for pointing out this terminology that needs to be clarified. We have now replaced “embryonic” with “prenatal” as a more appropriate description for the sampling time points in the manuscript.

1. *Drosophila* should be italicized in the introduction.

Thanks for suggesting that we have set the “*Drosophila*” words to italics in the manuscript.

1. Introduction - "In rodents, radial glia are found in the ventricular zone (VZ), where they undergo proliferation and differentiation." This sentence implies that only within rodents are radial glia found within the ventricular zone. Radial glia are present within the ventricular zone of all mammals.

Thanks for careful reading. This sentence has been corrected “In mammals, radial glial cells are found in the ventricular zone (VZ), where they undergo proliferation and differentiation.”

1. Figure 1A - an image of the E40 brain is missing.

We first sampled the prenatal developmental cortex of rhesus monkeys at the E40 timepoint. Unfortunately, we forgot to save the photo of the sampling at the E40 time point.

1. Figure 1B and 1C - it is unclear why cluster 20 is not annotated in Figure 1 as in the text it is stated "Each of the 28 identified clusters could be assigned to a cell type identity..." This cluster expresses VIM and PAX6 suggestive of ventricular radial glia and is located topographically approximate to IPC cluster 8 and seems to bridge the gap between neural stem cells and the interneuron clusters. Additionally, cluster 20 appears to be subclustered by itself in the progenitor subcluster UMAP (Figure 3A) suggestive of a batch effect or cells with low quality. The investigation, quality control, and proper annotation of this cluster 20 is necessary.

We appreciate for the reviewer’s suggestion. We detected specific expression marker genes of cluster 20, cells in this cluster specifically expressed VIM, IGFBP5 and TAC. According to the cell annotation results from a published study, we relabeled cluster 20 as ventral LGE-derived interneuron precursors (Yu, Yuan et al. Nat Neurosci. 2021. doi:10.1038/s41593-021-00940-3. PMID: 34737447.). Cluster 20 cells have been deleted in the new Figure 3A.

1. Figure 1B UMAP - it is unexpected that meningeal cells would cluster topographically closer to the excitatory neuron cluster (one could even argue that the meningeal cell cluster is located within the excitatory neuron clusters) instead of next to or with the endothelial cell clusters. This is suspicious for a mis-annotated cell cluster. ZIC2 and ZIC3 were used as the principal marker genes for meningeal cells. However, these genes are not specific for meninges (PanglaoDB) and had not been identified as marker genes in a developmental sc-RNAseq dataset of the developing mouse meninges (DeSisto et al. 2020). Additional marker genes (COL1A1, COL1A2, CEMIP, CYP1B1, SLC13A3) may be helpful to delineate the identity of this cluster and provide more evidence for a meningeal origin.

We thank the reviewer for the constructive advice. The violin plot in Author response image 3 has checked additional marker genes, including COL1A1, COL1A2, CEMIP, and CYP1B2. Cluster 23 does not express these marker genes but specifically expresses thalamus marker genes SHOX2(Rosin, Jessica M et al. Dev Biol. 2015. doi:10.1016/j.ydbio.2014.12.013. PMID: 25528224.) and TCF7L2(Lipiec, Marcin Andrzej et al. Development. 2020. doi: 10.1242/dev.190181. PMID: 32675279). According to the gene expression results, we corrected the cell definition of cluster 23 to thalamic cells in the revised manuscript. Specifically, we added marker genes SHOX2 and CYP1B1 in the new Figure 1D violin plot and corrected the cell definition of cluster23 from meninges to thalamus cells in the revised manuscript and figures.

**Author response image 3. sa4fig3:** Vlnplot of additional markers in cluster 23.

1. From Figure 1A, it appears that astrocytes (cluster 13) are present at E40 and E50 timepoints. This is inconsistent with literature and experimental data of the timing of the neuron-glia switch in primates and inconsistent with the claim within the text that, "Collectively, these results suggested that cortical neural progenitors undergo neurogenesis processes during the early stages of macaque embryonic cortical development, while gliogenic differentiation... occurs in later stages." The clarification of the percentage of astrocytes at each timepoint would clarify this point.

According to the suggestion, we have statistically analyzed the percentage of astrocytes (cluster 13) at each time point. The statistical results showed that the proportion of astrocytes was low to 0.1783% and 0.1046% at E40 and E50 time points, and increased significantly at E80 and E90, suggesting the onset of macaque gliogenesis might be around embryonic 80 days to 90 days. The result was consistent with published research on the timing of the neuron-glial transition in primates (Rash, Brian G et al. Proc Natl Acad Sci U S A. 2019. doi:10.1073/pnas.1822169116. PMID: 30894491). Besides, we thought that the cells in cluster 13 captured at E40 to E50 time points, with a total number of less than 200, maybe astrocyte precursor cells expressing the AQP4 gene (Yang, Lin, et al. Neuroscience bulletin. 2022. doi:10.1007/s12264-021-00759-9. PMID: 34374948).

1. A subcluster of ExN neurons was identified and determined to be of midbrain origin based on expression of TCF7L2. Did this subcluster express other known markers of the developing midbrain (OTX2, LMX1A, NR4A2, etc...)? Additionally, does this subcluster suggest that the limits of the dissection extended to the midbrain in samples E40 and E50?

We apologize for the previous inadequacy of the excitatory neuron cell annotation. In the description of the previous version of the manuscript, we misidentified the cells of the EN8 as midbrain cells. Following the reviewer’s suggestion, we verified the expression of more tissue- specific marker genes of EN8. As the violin diagram in Author response image 4 shows, other developing midbrain markers OTX2, NR4A2, and PAX7 did not express in EN8, but thalamus marker genes SHOX2, TCF7L2, and NTNG1 were highly expressed in EN8. Besides, dorsal cortex excitatory neuron markers NEUROD2, NEUROD6, and EMX1 were not expressed in EN8, which suggests that EN8 might not belong to cortical cells. After carefully reviewing the data analysis process, we determined that EN8 was a small group of cells in cluster 23 mistakenly selected during excitatory neuron analysis, as shown in Figure R5(A), which was corrected after revision. In the revised manuscript, we have removed EN8 from the analysis of excitatory neurons. In the revised manuscript, we have deleted the previous EN8 subcluster and renumbered the left excitatory neuron subclusters in new Figure 2 and Figure S3.

**Author response image 4. sa4fig4:** (A) Modified diagram of clustering of excitatory neuron subclusters collected at all time points, visualized via UMAP related to Figure 2A. (B) Vlnplot of different marker genes in EN8.

1. "These data suggested that the cell fate determination by diverse neural progenitors occurs in the embryonic stages of macaque cortical development and is controlled by several key transcriptional regulators" The authors present a list of differentially expressed genes specific to the various radial glia clusters along pseudotime. Some of these radial glia DEGs are known and have been characterized by previous literature while other DEGs they have identified had not been previously shown to be associated with radial glia specification/maturation. However, this list of DEGs does not support the claim that cell fate determination is controlled by several key transcriptional regulators. What were the transcriptional regulators of radial glia specification identified in this study and how were they validated?

We agree with the reviewer and honestly admit that the description of this part in the previous manuscript is inaccurate. The description has been deleted in the revised manuscrip.

1. "Comparing vRG to IPC trajectory between human, macaque, and mouse, we found this biological process of vRG-to-IPC is very conserved across species, but the vRG to oRG trajectory is divergent between species. The latter process is almost invisible in mice, but it is very similar in primates and macaque." Firstly, macaques are primates, and the text should be updated to reflect this. Secondly, from Figure 5C., it seems there were no outer radial glia detected at all within the vRG-oRG and vRG-IPC developmental trajectories. This would imply that oRGs are not "almost invisible" in mice, but rather do not exist. The authors need to clarify the presence or absence of identifiable outer radial glia in the integrated dataset and relate the relative abundance of these cells to their interpretation of the developmental trajectories for each species.

We apologize for the description inaccuracies in the manuscript and thank the reviewer for pointing out the expression errors. At your two suggestions, the description has been corrected in the revised manuscript as "Comparing vRG to IPC trajectory between human, macaque, and mouse, we found this biological process of vRG-to-IPC is very conserved across species. However, the vRG to oRG trajectory is divergent between species because the oRG population was not identified in the mouse dataset. The latter process is almost invisible in mice but similar in humans and macaques".

Although several published research has shown that oRG-like progenitor cells were present in the mouse embryonic neocortex(Wang, Xiaoqun et al. Nature neuroscience.2011. doi:10.1038/nn.2807; Vaid, Samir et al. Development. 2018, doi:10.1242/dev.169276. PMID: 30266827). However, oRG cells were barely detected in the scRNA-seq dataset of mice cortical development studies(Ruan, Xiangbin et al. Proc Natl Acad Sci U S A. 2021. doi:10.1073/pnas.2018866118. PMID: 33649223; Di Bella, Daniela J et al. Nature. 2021. doi:10.1038/s41586-021-03670-5. PMID: 34163074; Chen, Ao et al. Cell. 2022. doi:10.1016/j.cell.2022.04.003. PMID: 35512705). There were no oRG populations detected in the mouse embryonic cortical development dataset (GEO: GSE153164) used for integration analysis in our study.

1. "Ventral radial glia cells generate excitatory neurons by direct and indirect neurogenesis" This should be corrected to dorsal radial glia cells as this paper is discussing radial glia of the dorsal pallium.1. Editorially, gene names need to be italicized in the text, figures, and figure legends.1. Figure 5B - a scale bar showing the scale of the relative expression denoted by the dark blue color would be beneficial.1. Figure S7D is mislabeled in the figure legend.

Merged response to points 11 to 15: Thank you for kindly pointing out the errors in our manuscript. We have corrected the above four points in the revised version.

**Reviewer #2 (Recommendations For The Authors):**
Specific suggestions for authors:In the abstract the authors state: "thicker upper-layer neurons". I think it's important to be clear in the language by stating either that the layers are thicker or the neurons are most dense.

Thanks for your good comments. The description of “thicker upper-layer neurons” was corrected to “the thicker supragranular layer” in the revised manuscript. The supragranular layer thickness in primates was much higher than in rodents, both in absolute thickness and in proportion to the thickness of the whole neocortex (Hutsler, Jeffrey J et al. Brain research. 2005. doi:10.1016/j.brainres.2005.06.015. PMID: 16018988). Here, we want to describe the supragranular layer of primates as significantly higher than that of rodents, both in absolute thickness and in proportion to the thickness of the whole neocortex.

The introduction needs additional clarification regarding the vRG vs oRG discussion. I was unclear what the main takeaway for readers should be. Similarly, the discussion of previous studies and the importance for comparing human and macaque could be clarified.

We appreciate the suggestion and apologize for the shortcomings of the introduction part. We have rewritten the section and added additional clarification in the revised introduction. In the revised manuscript, the contents of the introduction are as follows:

“The neocortex is the center for higher brain functions, such as perception and decision-making. Therefore, the dissection of its developmental processes can be informative of the mechanisms responsible for these functions. Several studies have advanced our understanding of the neocortical development principles in different species, especially in mice. Generally, the dorsal neocortex can be anatomically divided into six layers of cells occupied by distinct neuronal cell types. The deep- layer neurons project to the thalamus (layer VI neurons) and subcortical areas (layer V neurons), while neurons occupying more superficial layers (upper-layer neurons) preferentially form intracortical projections1. The generation of distinct excitatory neuron cell types follows a temporal pattern in which early-born neurons migrate to deep layers (i.e., layers V and VI), while the later- born neurons migrate and surpass early-born neurons to occupy the upper layers (layers II-IV) 2. In *Drosophila*, several transcription factors are sequentially explicitly expressed in neural stem cells to control the specification of daughter neuron fates, while very few such transcription factors have been identified in mammals thus far. Using single-cell RNA sequencing (scRNA-seq), Telley and colleagues found that daughter neurons exhibit the same transcriptional profiles of their respective progenitor radial glia, although these apparently heritable expression patterns fade as neurons mature3. However, the temporal expression profiles of neural stem cells and the contribution of these specific temporal expression patterns in determining neuronal fate have yet to be wholly clarified in humans and non-human primates. Over the years, non-human primates (NHP) have been widely used in neuroscience research as mesoscale models of the human brain. Therefore, exploring the similarities and differences between NHP and human cortical neurogenesis could provide valuable insight into unique features during human neocortex development.

In mammals, radial glial cells are found in the ventricular zone (VZ), where they undergo proliferation and differentiation. The neocortex of primates exhibits an extra neurogenesis zone known as the outer subventricular zone (OSVZ), which is not present in rodents. As a result of evolution, the diversity of higher mammal cortical radial glia populations increases. Although ventricular radial glia (vRG) is also found in humans and non-human primates, the vast majority of radial glia in these higher species occupy the outer subventricular zone (OSVZ) and are therefore termed outer radial glia (oRG). Outer radial glial (oRG) cells retain basal processes but lack apical junctions 4 and divide in a process known as mitotic somal translocation, which differs from vRG 5. VRG and oRG are both accompanied by the expression of stem cell markers such as PAX6 and exhibit extensive self-renewal and proliferative capacities 6. However, despite functional similarities, they have distinct molecular phenotypes. Previous scRNA-seq analyses have identified several molecular markers, including HOPX for oRGs, CRYAB, and FBXO32 for vRGs7. Furthermore, oRGs are derived from vRGs, and vRGs exhibit obvious differences in numerous cell-extrinsic mechanisms, including activation of the FGF-MAPK cascade, SHH, PTEN/AKT, and PDGF pathways, and oxygen (O2) levels. These pathways and factors involve three broad cellular processes: vRG maintenance, spindle orientation, and cell adhesion/extracellular matrix production8.

Some transcription factors have been shown to participate in vRG generation, such as INSM and TRNP1. Moreover, the cell-intrinsic patterns of transcriptional regulation responsible for generating oRGs have not been characterized.

ScRNA-seq is a powerful tool for investigating developmental trajectories, defining cellular heterogeneity, and identifying novel cell subgroups9. Several groups have sampled prenatal mouse neocortex tissue for scRNA-seq 10,11, as well as discrete, discontinuous prenatal developmental stages in human and non-human primates 7,12 13,14. The diversity and features of primate cortical progenitors have been explored 4,6,7,15. The temporally divergent regulatory mechanisms that govern cortical neuronal diversification at the early postmitotic stage have also been focused on 16. Studies spanning the full embryonic neurogenic stage in the neocortex of humans and other primates are still lacking. Rhesus macaque and humans share multiple aspects of neurogenesis, and more importantly, the rhesus monkey and human brains share more similar gene expression patterns than the brains of mice and humans17-19. To establish a comprehensive, global picture of the neurogenic processes in the rhesus macaque neocortex, which can be informative of neocortex evolution in humans, we sampled neocortical tissue at five developmental stages (E40, E50, E70, E80, and E90) in rhesus macaque embryos, spanning the full neurogenesis period. Through strict quality control, cell type annotation, and lineage trajectory inference, we identified two broad transcriptomic programs responsible for the differentiation of deep-layer and upper-layer neurons. We also defined the temporal expression patterns of neural stem cells, including oRGs, vRGs, and IPs, and identified novel transcription factors involved in oRG generation. These findings can substantially enhance our understanding of neocortical development and evolution in primates.”

Why is this study focused on the parietal lobe? This should be discussed in the introduction and interpretation of the data should be contextualized in the context of this cortical area.

In this study, samples were collected from the parietal lobe area mainly for the following reasons:

(1) To ensure that the cortical anatomical parts collected at each time point are consistent, we used the lateral cerebral sulcus as a marker to collect the parietal lobe tissue above the lateral sulcus for single-cell sequencing sample collection. Besides, the parietal region is also convenient for sampling the dorsal cortex.

(2) Previous studies have made the timeline of the macaque parietal lobe formation process during the prenatal development stage clear （ Finlay, B L, and R B Darlington.Science.1995. doi:10.1126/science.7777856. PMID: 7777856）, which is also an essential reason for using the parietal lobe as the research object.

Figure 1:Difficult to appreciate how single cell expression reflects the characterization of layers described in Figure 1A. A schematic for temporal development would be helpful. Also, how clusters correspond to discrete populations of excitatory neurons and progenitors would improve figure clarity. Perhaps enlarge and annotate the UMAPS on the bottom of Figure 1A.

We thank the reviewer for the suggestion and apologize for that Figure 1A does not convey the relationship between single-cell expression and neocortex layer formation. In the revised manuscript, time points information associated with the hierarchy is labeled to the diagram in Figure S1A. The UMAPS on the bottom of Figure 1A was enlarged in the revised manuscript as new Figure 1C.

Labels on top of clusters for 1A/1B would be helpful as it's difficult to see which colors the numbers correspond to on the actual UMAP.

Many thanks to the reviewer for carefully reading and helpful suggestions. We have adjusted the visualization of UMAP in the revised vision. The numbers in the label bar of Figure 1B have been moved to the side of the dot so that the dot can be seen more clearly.

Microglia and meninges are also non-neural cells. This needs to be changed in the discussion of the results.

Thanks for the suggestion. We have fixed the manuscript as the reviewer suggested. The description in the revised manuscript has been fixed as follows: “According to the expression of the marker genes, we assigned clusters to cell type identities of neurocytes including radial glia (RG), outer radial glia (oRG), intermediate progenitor cells (IPCs), ventral precursor cells (VP), excitatory neurons (EN), inhibitory neurons (IN), oligodendrocyte progenitor cells (OPC), oligodendrocytes, astrocytes, ventral LGE-derived interneuron precursors and Cajal-Retzius cells, or non-neuronal cell types including microglia, endothelial, meninge/VALC(vascular cell)/pericyte, and blood cells. Based on the expression of the marker gene, cluster 23 was identified as thalamic cells, which are small numbers of non-cortical cells captured in the sample collection at earlier time points. Each cell cluster was composed of multiple embryo samples, and the samples from similar stages generally harbored similar distributions of cell types.”.

It's important to define the onset of gliogenesis in the text and figure. What panels/ages show this?

We identified the onset of gliogenesis by statistically analyzing the percentage of astrocytes (cluster 13) at each time point and added the result in Figure S1. The statistical results showed that the proportion of astrocytes was deficient at E40 and E50 time points and increased significantly at E80 and E90, suggesting the onset of macaque gliogenesis might be around embryonic 80 days to 90 days. The result was consistent with published research on the timing of the neuron-glial transition in primates (Rash, Brian G et al. Proceedings of the National Academy of Sciences of the United States of America 201. doi:10.1073/pnas.1822169116. PMID: 30894491).

Figure 2:Why are there so few neurons at E90? Is it capture bias, dissociation challenges (as postulated for certain neuronal subtypes in the discussion), or programmed cell death at this time point?

We thought it was because mature neurons at E90 with abundant axons and processes were hard to settle into micropores of the BD method for single cell capture. Due to the fixed size of the BD Rhapsody microwells, this sing-cell capture method might be less efficient in capturing mature excitatory neurons but has a good capture effect on newborn neurons at each sampling time point. In conclusion, based on the BD cell capture method feature, the immature neurons at each point are more easily captured than mature neurons in our study, so the generation of excitatory neurons at different developmental time points can be well observed, as shown in Figure 2, which aligns with our research purpose.

The authors state: "We then characterized temporal changes in the composition of each EN subcluster. While the EN 5 and EN 11 (deep-layer neurons) subclusters emerged at E40 and E50 and disappeared in later stages, EN subclusters 1, 2, 3, and 4 gradually increased in population size from E50 to E80 (Figure 2D)." What about EN7? It's labeled as an upper layer neuron that is proportionally highest at E40. Could this be an interesting, novel finding? Does this indicate something unique about macaque corticogenesis? The authors don't describe/discuss this cell type at all.

We apologize for the manuscript’s lack of detailed descriptions of EN results. In our study, EN7 is identified as CUX1-positive, PBX3-positive, and ZFHX3-positive excitatory neuron subcluster. The results of Fig. 2B show that EN7 was mainly captured from the early time points (E40/E50) samples. Above description was added in the revised manuscript.

The Pbx/Zfhx3-positive excitatory neuron subtype reported in Moreau et al. study on mouse neocortex development progress （ Moreau, Matthieu X et al. Development. 2021. doi:10.1242/dev.197962. PMID: 34170322）. Our study verified that the Pbx3/Zfhx3-positive cortical excitatory neurons also exist in the early stage of prenatal macaque cortex development.

Is there any unique gene expression in identified subtypes that are surprising? Did the comparison against human data, in later figures, inform any unique features of gene expression?

Based on the excitatory neuron subclusters analysis result in our study, we found no astonishing results in excitatory neuron subclusters. In subsequent integrated cross-species analyses, macaque excitatory neurons showed similar transcriptional characteristics to human excitatory neurons. In general, excitatory neurons tend to have a greater diversity in the cortex of animals that are more advanced in evolution (Ma, Shaojie et al. Science. 2022. doi:10.1126/science.abo7257. PMID: 36007006; Wei, Jia-Ru et al. Nat Commun. 2022. doi:10.1038/s41467-022-34590-1. PMID:36371428; Galakhova, A A et al. Trends Cogn Sci. 2022. doi:10.1016/j.tics.2022.08.012. PMID: 36117080; Berg, Jim et al. Nature. 2021. doi:10.1038/s41586-021-03813-8. PMID: 34616067).Since only single-cell transcriptome data was analyzed in this study, we did not find any unique features of the prenatal developing macaque cortex excitatory neurons in the comparison against the human dataset due to the limitation of information dimension.

Figure 3:The identification of terminal oRG differentiation genes is interesting. The confirmation of known gene expression as well as novel markers that indicate different states/stages of oRG cells is a valuable resource. As the identification of described ion channel expression is a novel finding, it should be explored more and would be strengthened by validation in tissue samples and, if possible, functional assays.E is the most novel part of this figure, but it's very hard to read. I think increasing the focus of this figure onto this finding and parsing these results more would be informative.

Thanks for the positive comments. We apologize for the lack of clarity and conciseness in figure visualizations. We hypothesized vRG to oRG cell trajectories into three phases: onset, commitment, and terminal. The leading information conveyed by Figure 3E was the dynamic gene expression along the developmental trajectory from vRG to oRG. Specific genes were selected and shown in the schema diagram of new Figure 3.

We verified terminal oRG differentiation genes in the recently published macaque telencephalic development dataset(Micali N, Ma S, Li M, et al. Science. doi:10.1126/science.adf3786.PMID: 37824652) (GEO accession: GSE226451). The results of Author response image 2 show that the gene expression showed states/stages. Most of the oRG terminal differentiation markers genes identified in our study were also expressed in the oRG cells of the GSE226451 dataset. In particular, the two datasets were consistent in the expression of ion channel genes ATP1A2, ATP1A2, and SCN4B.

I'm curious about the granularity of the oRG_C12 terminal cluster. Are there ways to subdivide the different cells that seem to be glial-committed vs actively dividing vs neurogenically committed to IPCs? In the text, the authors referred to different oRG populations, but they are annotated as the same cluster and cell type. The authors should clarify this.

According to the reviewer's suggestion, we subdivide the oRG_C12 into eight subclusters. Based on the marker gene in Author response image 5C, subclusters 1,2 and 4 might be glial- committed with AQP4/S100B positive expression; subclusters 3,6,7 might be neurogenically committed to IPCs with NEUROD6 positive expression; subclusters 0,3,5,6,7 might be actively dividing state with MKI67/TOP2A positive expression.

**Author response image 5. sa4fig5:** Subdivide analysis of oRG_C12. (A)and (B) Subdividing of e oRG_C12 visualized via UMAP. Cells are colored according to subcluster timepoint (A) and subcluster identities (B). (C) Violin plot of molecular markers for the subclusters.

Figure 4:Annotating/labeling the various EN clusters (even as deep/upper) would help improve the clarity of this and other figures. It's clear what each progenitor subtype is but it's hard to read the transitions. Why are all the EN groups in pink/red? It makes the data challenging to interpret.

In Figure4A, we use different yellow/orange colors for deep-layer excitatory neuron subclusters (EN5 and EN10), and different red/pink colors for upper-layer excitatory neuron subclusters (EN1, EN2, EN3, EN4, EN6, EN7, EN8 and EN9). We add the above information in the legend of Figure 4 in the revised manuscript.

E50 seems to be unique - what's EN11?

Based on the molecular markers for EN subclusters in Author response image 2, we recognized EN11 as a deep-layer excitatory neuron subcluster expressing BCL11B and FEZF2. As explained in the above reply, the microplate of BD has a good effect on capturing newborn neurons at each time point. The EN11 was mainly a newborn excitatory neuron at the E50 timepoint, which makes the subcluster seem unique.

**Author response image 6. sa4fig6:** Vlnplot of different markers in EN8.

Figure 4E - the specificity of gene expression for deep vs upper layer markers seems to be over stated given the visualized gene expression pattern (ex FEZF2). Could the right hand panels be increased to better appreciate the data and confirm the specificity, as described.

In our study, we used slingshot method to infer cell lineages and pseudotimes, which have been used to identifying biological signal for different branching trajectories in many scRNA- seq studies. We apologize for the lack of visualization clarity in the figure 4E. Due to the size limitation of the uploaded file, the file was compressed, resulting in a decrease in the clarity of the image. Below, we provided figure 4E with a higher definition and increased several genes’ slingshot branching tree results according to the reviewer's suggestion.

Figure 5:There are some grammatical typos at the bottom of page 8. In this section, it also feels like there is a missing logical step between expansion of progenitors through elongated developmental windows that impact long-term expansion of the upper cortical layers.

We apologize for the grammatical typos and have corrected them in the revised manuscript. We understand the reviewer’s concern. Primates have much longer gestation than rodents, and previous study evidence had shown that extending neurogenesis by transplanting mouse embryos to a rat mother increases explicitly the number of upper-layer cortical neurons, with concomitant abundant neurogenic progenitors in the subventricular zone(Stepien, Barbara K et al. Curr Biol. 2020. doi:10.1016/j.cub.2020.08.046. PMID: 32888487). We thought this mechanism could also explain primates' much more expanded abundance of upper-layer neurons.

I'm curious about the IPCs that arise from the oRGs. Lineage trajectory shows vRG decision to oRG or IPC, but oRGs also differentiate into IPCs. Could the authors conjecture why they are not in this dataset or are indistinguishable from vRG-derived IPCs.

Several published experiments have proved that oRG can generate IPC in human and macaque developing neocortex. (Hansen, David V et al. Nature. 2010. doi:10.1038/nature08845. PMID: 20154730; Betizeau, Marion et al. Neuron. 2013. doi:10.1016/j.neuron.2013.09.032. PMID: 24139044). Clearly identifying the difference between IPC generated from vRG and oRG at the transcriptional level in our single-cell transcriptome dataset is difficult. We hypothesized that the IPCs produced by both pathways have highly similar transcriptional features. Due to the limit of the scRNA data analysis algorithm used in this study, we didn’t distinguish the two kinds of IPC, which could not be in terms of pseudo-time trajectory reconstruction and transcriptional data.

Figure 6 :How are the types 1-5 in 6A defined? Were they defined in one species and then applied across the others?

We applied the same analysis to each species. We first picked up vRG cells in each species dataset and screened the differentially expressed genes (DEGs) between adjacent development time points using the “FindMarkers” function (with min. pct = 0.25, logfc. threshold = 0.25). After separate normalization of the DEG expression matrix from different species datasets, we use the “standardise” function from the Mfuzz package to standardize the data. The DEGs of vRG in each species were grouped into five clusters using the Mfuzz package in R with fuzzy c- means algorithm.

The temporal dynamics in the highlighted section in B have interesting, consistent patterns of gene expression of the genes described, but what about the genes below that appear less consistent temporally? What processes do not appear to be conserved, given those gene expression differences?

Many thanks for the constructive comments. The genes in Figure 6B below are temporal dynamics non-conserved transcription factors among the three species vRG. We performed a functional enrichment analysis on the temporal dynamics of non-conserved transcription factors with the PANTHER (Protein ANalysis THrough Evolutionary Relationships) Classification System(https://www.pantherdb.org/), and the analysis results are shown in Author response image 7. The gene ontology (GO) analysis results show that unconserved transcription factors were related to different biological processes, cellular components, and molecular functions. However, subsequent experiments are still needed to verify specific genes.

**Author response image 7. sa4fig7:** Gene Ontology (GO) analysis of unconserved temporal patterns transcription factors among mouse, macaque and human vRG cells.

The identification of distinct regulation of gene networks, despite conservation of transcription factors in discrete cell types, is interesting. What does the comparison between humans and macaques indicate about regulatory differences evolutionarily?

We appreciate the reviewer for the comments. We performed the TFs regulation network analysis of human vRG with pyscenic workflow. The top transcription factors of every time point in human vRG were calculated, and we used the top 10 TFs and their top 5 target genes to perform interaction analysis and generate the regulation network of human vRG in revised figure 6. In comparison of the pyscenic results of mouse, macaque and human vRG, it was obvious that the regulatory networks were not evolutionarily conservative. Compared with macaque, the regulatory network of transcription factors and target genes in humans is more complex. Some conserved regulatory relationships present in more than one species are identified, such as HMGN3, EMX2, SOX2, and HMGA2 network at an early stage when deep lager generation and SOX10, ZNF672, ZNF672 network at a late stage when upper-layer generation.

**Reviewer #3 (Recommendations For The Authors):**
The data should be compared to a similar brain region in human and mouse, if available. (See data from PMCID: PMC8494648).

We appreciate the reviewer’s suggestions. In Figure 6, the species-integration analysis, the mouse data were from the perspective of the somatosensory cortex, macaque data were mainly from the parietal lobe in this study, and human data including the frontal lobe (FL), parietal lobe (PL), occipital lobe (OL), and temporal lobe (TL). PMC8494648 offered high-quality data covering the period of gestation week 14 to gestation week 25. However, our study's development stage of rhesus monkeys is E40-E90 days, corresponding to pcw8-pcw21 in humans. The quality of data from PMC8494648 is particularly good. However, the developmental processes covered by PMC8494648 don’t perfectly match the development time of the macaque cortex that we focused on in this study. Therefore, it is challenging to integrate the dataset (PMCID: PMC8494648) into the data analysis part. However, we have cited the results of this precious research (PMCID: PMC8494648) in the discussion part of the revised manuscript.

A deeper assessment of these data in the context of existing studies would help distinguish the work and enable others to appreciate the significance of the work.

We appreciate the reviewer’s constructive suggestions. The human regulation analysis with pyscenic workflow was added into new figure 6 for the comparison of different species vRG regulatory network. Analysis of the regulatory activity of human, macaque and mouse prenatal neocortical neurogenesis indicated that despite commonalities in the roles of classical developmental TFs such as GATA1, SOX2, HMGN3, TCF7L1, ZFX, EMX2, SOX10, NEUROG1,NEUROD1 and POU3F1. The top 10 TFs of the human, macaque, and mouse vRG each time point and their top 5 target genes identified by pySCENIC as an input to construct the transcriptional regulation network (Figure 6 D, F and H). Some conserved regulatory TFs present in more than one species are identified, such as HMGN3, EMX2, SOX2, and HMGA2 at an early stage when deep- lager generation and SOX10, ZNF672, and ZNF672 at a late stage when upper-lay generation.

Besides, we performed some comparative analysis with our macaque dataset and the newly published macaque telencephalon development dataset. The results were only used to provide additional information to reviewers and were not included in the revised manuscript.

To verify the reliability of our cell annotation results, we compared the similarity of cell-type association between our study and recently published research(Micali N, Ma S, Li M, et al. Science. doi:10.1126/science.adf3786.PMID: 37824652), using the scmap package to project major cell types in our macaque development scRNA-seq dataset to GSE226451. The river plot in Author response image 1 illustrates the broadly similar relationships of cell type classification between the two datasets. Otherwise, we used more marker genes for cell annotation to improve the results of cell type definition in new Figure 1D. Besides, the description of distinct excitatory neuronal types has been improved in the new Figure 2.

Furthermore, we verified terminal oRG differentiation genes in the recently published macaque telencephalic development dataset(Micali N, Ma S, Li M, et al. Science. doi:10.1126/science.adf3786.PMID: 37824652) (GEO accession: GSE226451). The results of Authro response image 2 show that the gene expression showed states/stages. Most of the oRG terminal differentiation markers genes identified in our study were also expressed in the oRG cells of the GSE226451 dataset. In particular, the two datasets were consistent in the expression of ion channel genes ATP1A2, ATP1A2, and SCN4B.